# Heterogeneity of IL-15-expressing mesenchymal stromal cells controls natural killer cell development and immune cell homeostasis

Carmen Stecher [1], Romana Bischl[1], Anna Schmid-Böse [1], Stefanie Ferstl[1], Elisabeth Potzmann[1], Magdalena Frank [1], Nina Braun[1], Matthias Farlik [2], Richard A. Flavell [3] & Dietmar Herndler-Brandstetter [1] ✉

Bone marrow (BM) mesenchymal stromal cells (MSC) provide microenvironmental niches that support hematopoietic stem cells and regulate hematopoiesis. Whether functional heterogeneity among BM MSCs contributes to the development and survival of distinct immune cell lineages remains incompletely understood. Here, we use an *Il15* knockin reporter and multiple conditional deletion mouse models to show distinct differences in IL-15 expression between BM MSC subtypes. Conditional deletion of *Il15* in *Osx*+ stromal cells results in decreased natural killer (NK) cell precursors, memory CD8+ T cells and NKT cells but not mature NK cells. *Lepr*+ stromal cells support the survival of mature NK cells and memory CD8+ T cells in the BM of older mice, while endothelial cells support mature NK cells and memory CD8+ T cells in the blood but not in the BM. Thus, our data suggest that MSC subtypes differentially regulate the development and survival of IL-15-dependent immune cell lineages in the BM.

The bone marrow (BM) stroma regulates hematopoiesis and provides specialized microenvironmental niches for the long-term survival of hematopoietic stem cells (HSC) and immunological memory. Stromal cells produce extracellular matrix to promote organ structure, but also express cytokines, chemokines and a distinct set of receptors, thereby promoting cell–cell interactions and cell positioning essential for hematopoiesis[1]. Some of these lineage-instructive signals are the C-X-C motif chemokine ligand 12 (CXCL12), stem cell factor (SCF), interleukin (IL)-6, IL-7 and IL-15[2–4]. HSCs, for example, reside and self-renew in endothelial and perivascular niches formed by mesenchymal progenitors and endothelial cells (EC) that produce CXCL12 and SCF[5–8]. Although CXCL12+ SCF+ BM mesenchymal stromal cells (MSC; including mesenchymal stem and progenitor cells as well as differentiated stromal cells) that support HSCs have been studied in great detail[5–7]

and recent single-cell sequencing results indicate transcriptional heterogeneity among mouse BM MSCs[9–11], the molecular and functional heterogeneity of interleukin 15 (IL-15) (co-)expressing MSC subsets remains less well understood.

IL-15 has been implicated in natural killer (NK) cell development[12] and type 1 innate lymphoid cell (ILC1) function[13] as well as memory CD8+ T cell[14–17] and NKT cell[18] survival. Hematopoietic cells, in particular macrophages and dendritic cells, have long been considered to be the major source of IL-15 in the BM[14,19]. However, several studies have suggested non-hematopoietic cells as a source of IL-15[19–22] and combined deletion of IL-15Rα from macrophages and dendritic cells did not affect NK cell numbers in the BM[14]. In vitro experiments further indicate that stromal cells are required for NK cell development in the absence of exogenous IL-15[23], and CXCL12-CXCR4 signaling is essential

---

[1]Center for Cancer Research, Medical University of Vienna and Comprehensive Cancer Center, Vienna, Austria. [2]Department of Dermatology, Medical University of Vienna and Comprehensive Cancer Center, Vienna, Austria. [3]Department of Immunobiology, Howard Hughes Medical Institute, Howard Hughes Medical Institute, New Haven, CT, USA. ✉e-mail: dietmar.herndler-brandstetter@meduniwien.ac.at

for NK cell development in adult mice[24]. These findings are in agreement with a study that used *Il15-CFP* knockin/knockout mice and identified IL-15[+] BM stromal cells with a VCAM-1[+] PDGFRβ[high] CD31[-] Sca-1[-] phenotype[20]. Together, these studies indicate that IL-15-expressing MSC subsets may provide unique niches essential for NK cell lineage specification, maturation and survival[25,26].

In addition to its role as a primary lymphoid organ, the BM is a major reservoir for memory CD8[+] T cells[15]. DC-derived IL-15 supports central-memory CD8[+] T cells (CD8[+] $T_{CM}$), whereas macrophage-derived IL-15 supports CD8[+] $T_{CM}$ and effector-memory (CD8[+] $T_{EM}$) cells in the BM[14]. However, combined deletion of IL-15Rα from macrophages and DCs only leads to a 50% reduction of memory CD8[+] T cells in the BM compared to complete IL-15 deletion[14]. The role of IL-15[+] stromal cell subsets in providing distinct signals for positioning, long-term survival and tissue-residency of memory CD8[+] T cell subsets in the BM has not been addressed so far[15,17,27,28].

In this study, we generate a bicistronic *Il15-IRES-EGFP* knockin reporter mouse model to faithfully identify and characterize *Il15*-expressing stromal cells in the BM. Single-cell transcriptomics and flow cytometry reveal heterogeneity among stromal cell subsets to produce *Il15*, with the highest frequency of *Il15*[+] stromal cells among LepR[+] VCAM-1[+] MSCs and ECs. Conditional deletion of *Il15* in *Osx*[+] stromal cells impairs NK cell development, NKT cells, CD8[+] $T_{CM}$ and $T_{RM}$ cells in the BM, whereas inducible conditional deletion of *Il15* in ECs impairs mature NK (mNK) cells and CD8[+] $T_{CM}$ in the blood but not in the BM. Our results reveal heterogeneity of *Il15*[+] stromal cells in the BM, which governs functional specialization to provide contextualized, non-redundant signals required to control distinct steps in NK cell development as well as the survival of NKT cells, CD8[+] $T_{CM}$ and $T_{RM}$ cells but not mNK cells or iILC1.

## Results

### Single-cell transcriptomic profiling of *Il15*-expressing stromal cells in the BM

To explore the heterogeneity of *Il15* expression in stromal cells in the BM, we first integrated droplet-based scRNA-seq datasets from four studies[9,10,29,30], generating a single cell transcriptomic atlas of 68.616 BM stromal cells (Fig. 1A, Table S3). The UMAP projection depicts 11 clusters of stromal cells in the BM, with all clusters being present, yet differentially abundant, in all of the individual integrated scRNA-seq datasets. The feature plots show key marker genes expressed in the different stromal cell clusters, such as leptin receptor (LepR; MSC_stem), cadherin 5 (*Cdh5*; ECs), Sp7 transcription factor 7 (*Osx*; MSC_osteo and chondrocytes) and regulator of G-protein signaling 5 (Rgs5; pericytes) (Fig. 1B). *Il15* is expressed in several stromal cell clusters, with the highest frequency of *Il15*[+] cells in MSC_stem, MSC_chondro and LepR[+] ECs (Fig. 1C).

However, scRNA-seq analysis may underestimate the frequency of *Il15*-expressing stromal cells, in particular those cells that display a low expression of *Il15* mRNA. To faithfully identify and purify Il15 protein expressing stromal cells, we therefore generated an *Il15* reporter knockin mouse model by targeted insertion of an *IRES-EGFP* construct after the translation stop codon of the mouse *Il15* gene (Fig. 1D). *Il15* mRNA was highly expressed in purified *Il15*[GFP+] CD45[-] Ter119[-] BM stromal cells, whereas it was absent in *Il15*[GFP-] BM stromal cells. Similarly, *Il15*[GFP+] BM stromal cells showed a trend of *Il15ra* co-expression (Fig. S1A). Further validation of our reporter mouse demonstrated *Il15* expression in CD8[+] dendritic cells (DC) and Ly6C[hi] macrophages, while *Il15* expression was absent in CD8[-] DCs, CD4[+] and CD8[+] T cells (Fig. S1B), and qPCR of sorted *Il15*[GFP-] and *Il15*[GFP+] BM macrophages confirmed the absence of *Il15* transcripts in *Il15*[GFP-] cells (Fig. S1C). This is in agreement with previous studies analyzing IL-15 expression[20,31]. Next, we sorted *Il15*[GFP+] CD45[-]Ter119[-]CD71[-] stromal cells from the BM of *Il15*[GFP] mice (Fig. 1E) and performed scRNA-seq, generating a single cell transcriptomic atlas of 6.943 *Il15*[GFP+] BM stromal cells, which were

annotated with the same projected cluster labels as in Fig. 1A (Fig. 1F). The feature plots show cluster-defining expression of genes, with *Lepr*[+], *Cdh5*[+] and *Sp7*[+] clusters being the most abundant ones in purified *Il15*[GFP+] BM stromal cells (Fig. 1G). Although our sort resulted in a high purity of *Il15*[GFP+] stromal cells (Fig. S1D), scRNA-seq was able to detect *Il15* mRNA in only 51% of MSC_stem and 29% of ECs (Fig. 1G). The use of purified *Il15*[GFP+] BM stromal cells therefore allowed for a more comprehensive analysis of a high number of *Il15*[+] BM stromal cells, including cells with a low abundance of *Il15* mRNA, as observed in ECs and some osteo-lineage MSCs (Fig. 1C). The heat map displays cluster signature genes (top 10 differentially expressed genes between predicted clusters) in *Il15*[GFP+] BM stromal cells (Fig. 1H) and the relative abundance of *Il15*-expressing BM stromal cells in the published and *Il15*[GFP+] scRNA-seq dataset is shown in Fig. 1I. Unsupervised clustering of *Il15*[GFP+] BM stromal cells revealed eight similarly distributed clusters; the top 5 differentially expressed genes of each cluster are shown in Fig. S1E. In order to directly compare *Il15* expression between sorted cells and total BM stroma, we then integrated and clustered our IL15[GFP+] cells with the public datasets to a meta-dataset using the same quality control parameters and gene annotations, showing *Il15* enrichment in *Il15*[GFP+] BM stromal cells (Fig. S1F–H). Taken together, our results demonstrate a distinct pattern of *Il15* expression intensity and frequency among stromal cell subtypes in the BM.

### Frequency and marker expression of *Il15-GFP*[+] stromal cells in the BM

To analyze the abundance and markers on *Il15*[GFP+] BM stromal cells, we performed a series of flow cytometry analyses. 65% of the LepR[+] VCAM-1[+] Lineage[-] CD31[-] MSC_stem population expressed *Il15*[GFP] (Figs. 2A and S2A). Most CXCL12[+] and 26.9% of *Prx1*-Tomato[+] stromal cells in the BM expressed *Il15*[GFP] (Figs. 2B and S2B). About 25% of CD200[+] CD24[+] hypertrophic chondrocytes expressed *Il15*[GFP], whereas *Il15*[GFP] expression was absent in CD200[+] CD24[-] proliferating chondrocytes (Figs. 2C and S2C). In contrast to scRNA-seq datasets, about 50% of sinusoidal ECs expressed *Il15*[GFP], whereas only few arteriolar ECs expressed *Il15*[GFP] (Figs. 2D and S2D). When comparing *Il15*[GFP+] and *Il15*[GFP-] Lin[-] CD45[-] CD31[-] stromal cells in the BM, *Il15*[GFP+] cells were expressing Cxcl12[DsRed], Prx1[Tomato], CD51, CD73 and VCAM-1. A high proportion also expressed CD200 and LepR but fewer *Il15*[GFP+] than *Il15*[GFP-] cells expressed CD24 and CD44 (Fig. 2E). *Il15*[GFP+] vs. *Il15*[GFP-] cells were enriched in skeletal stem cell markers (Ctsk, Ddr2) as well as osteo-lineage markers, such as Sp7 (Osx), Bmp6 and CD51 (Fig. S2E, F).

### Deletion of IL-15 in *Prx1*-expressing stromal cells impairs NK cell precursors and memory CD8[+] T cells in the BM

Studies with *Il15* and *Il15ra* total knockout mice have demonstrated a role for IL-15 in NK cell development as well as survival of mNK cells, memory CD8[+] T cells and NKT cells[12,32]. However, combined deletion of *Il15ra* in macrophages and DCs did not affect NK cell precursors or mNK cells in the BM and reduced BM memory CD8[+] T cells by only half[14]. Because our scRNA-seq and flow cytometry results demonstrated that stromal cells represented a significant source of IL-15 in the BM, we systematically analyzed the in vivo impact of IL-15-expressing stromal cell subtypes on different immune cell lineages using conditional *Il15* knockout mouse models.

We first deleted IL-15 from multipotent mesenchymal progenitor cells using *Il15*[flox/flox] mice (generated and validated by Nan-Shin Liao[21]) crossed with *Prx1-Cre* mice. *Prrx1* is a transcription factor expressed in the limb bud mesoderm[33] and deleted in all mesenchymal stromal cells[7,34]. Our results demonstrate a significant decrease of NKG2D[+] rNKPs and stage A immature NK (iNK) cells in the BM, whereas LSK, CLPs, rNKPs, stage B/C iNK cells and CD49b[+] mNK cells were not affected (Figs. 3A–D and S3A–D). CD44[hi] CD62L[+] central-memory CD8[+] T cells ($T_{CM}$) were reduced in the BM and spleen but not in the blood, and CXCR3[+] CD8[+] T cells were slightly reduced in the BM (Figs. 3E, F, S3E,

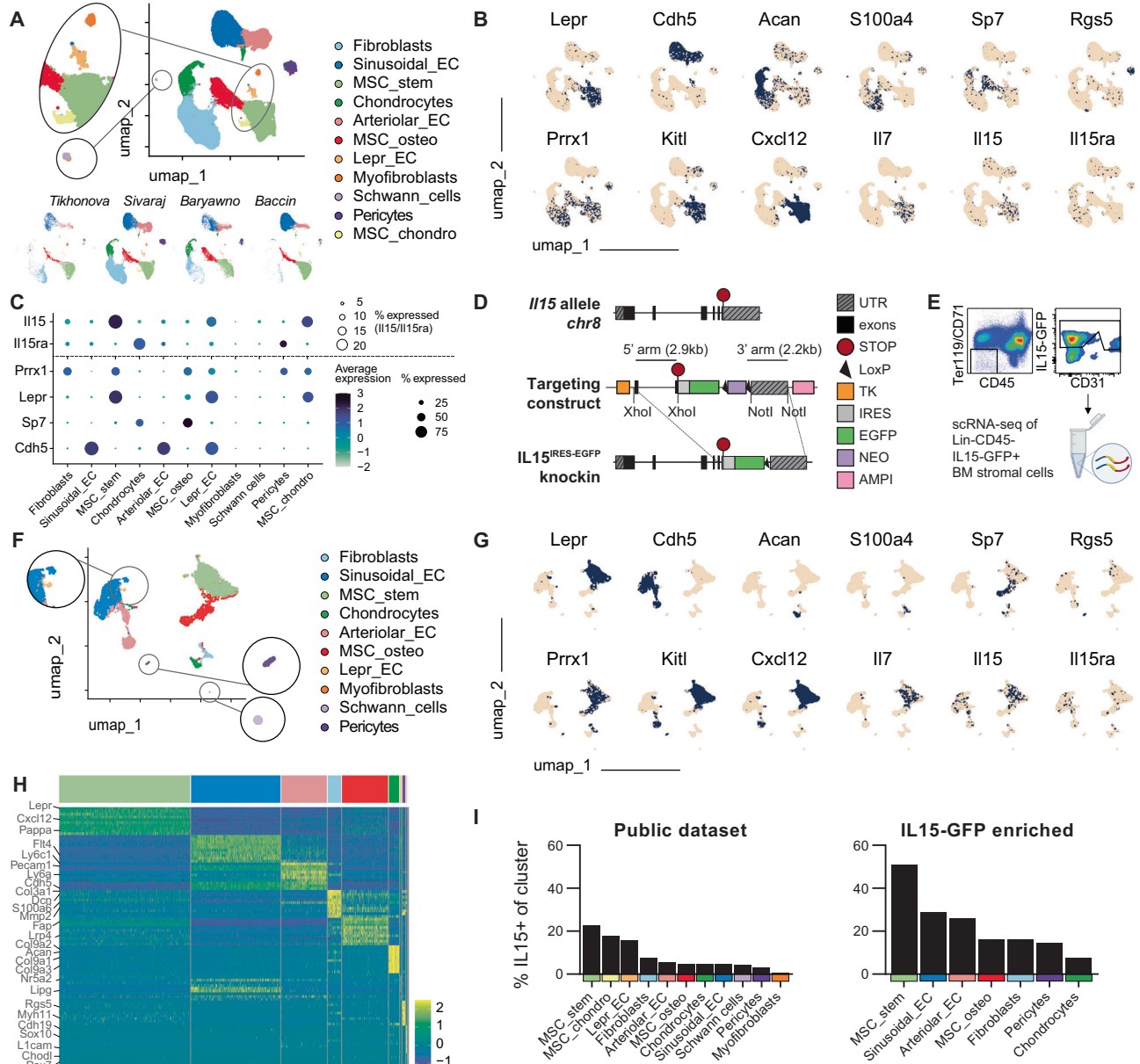

**Fig. 1 | Single-cell transcriptomic atlas of *Il15*-expressing stromal cells in the BM. A** UMAP projection of 68.616 BM stromal cells integrated from the four indicated published datasets. **B** Feature plots showing key marker genes expressed by the different clusters. **C** Dotplot showing selected differential gene expression by stromal cell clusters identified in the BM. Dot size indicates the abundance of *Il15/Il15ra* (top) or *Cre*-targeted genes (bottom). **D** Vector map of the generated *Il15GFP* knockin mice. **E** Flow cytometric gating strategy for the FACS-enrichment of *Il15*-expressing stromal cells from *Il15GFP* mouse BM. Created in BioRender. Lab Account, D. (2025) https://BioRender.com/wc1m6su. **F** UMAP projection of 6.943 *Il15GFP+* stromal cells sorted from the bone and BM of a total of seven *Il15GFP* mice. The cluster labels were projected based on the labels of the integrated public dataset in **A** by using Seurat's FindTransferAnchors function. **G** Feature plots showing cluster-defining expression of genes in *Il15GFP+* BM stromal cells. **H** Heat map showing top 10 differentially expressed genes between projected labels in the *Il15GFP+* dataset. **I** Relative abundance of *Il15*-expressing BM stromal cells in the published and *Il15GFP+* scRNA-seq datasets. Only clusters with more than 30 total cells are depicted.

S4A–D). In contrast, tissue-resident memory CD8$^+$ T cells (T$_{RM}$), NKT cells and immature type 1 innate lymphoid cells (iILC1), as well as CD4$^+$ T$_{CM}$, which depend on IL-7, were not affected (Figs. 3E, F, and S4B–D). While the loss of NK cell precursors was apparently compensated during the steady state in later stages of NK cell differentiation, we next investigated whether this phenotype would be exacerbated in times of heightened demand for NK cell differentiation. Therefore, we used an antibody-mediated NK cell depletion model, which showed that *Il15flox/flox* *Prx1-Cre* mice could not cope with the heightened demand of rNKPs (Fig. 3G). Interestingly, the CD27$^-$CD11b$^+$ mature NK cell population was also significantly reduced in Cre mice (Fig. S5A), similar to KLRG1-expressing NK cells (Fig. S5B). While NK cell proliferation during the steady state was similar in WT and Cre mice, *Prx1-Cre* mice had a slightly higher MFI of the anti-apoptotic protein Bcl-2 (Fig. S5C, D). However, NK cells from *Prx1-Cre* mice were not functionally impaired since they had a similar capacity to produce IFN-γ and TNF-α upon stimulation (Fig. S5E). Together, these results indicate that *Prrx1$^+$* stromal cells support NK cell development and survival of CD8$^+$ T$_{CM}$ cells in the BM but do not affect mNK, CD8$^+$ T$_{RM}$, NKT cells and iILC1 in the BM.

**Deletion of IL-15 in osteo-lineage stromal cells impairs NK cell precursors, NKT cells and memory CD8$^+$ T cells in the BM**
Because IL-15$^+$ BM stromal cells displayed an enrichment of osteogenic lineage genes, we deleted IL-15 from osteoblasts, osteocytes and

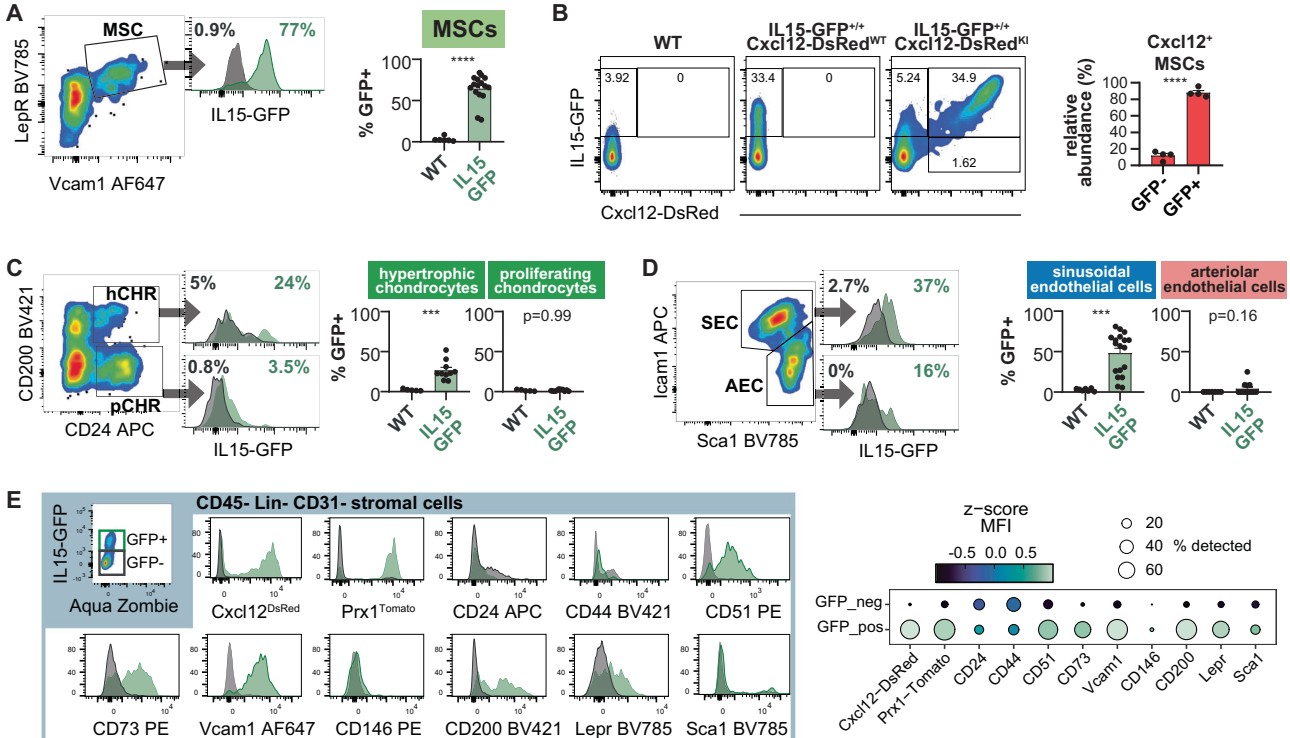

**Fig. 2 | Characteristics of Il15-expressing stromal cells in the BM. A** Flow cytometry staining of in *Il15^GFP+/+* mice (green) or WT controls (grey histograms) pre-gated on CD45⁻CD31⁻Lin⁻(TER-119/CD71) showing a high amount of LepR⁺VCAM-1⁺ MSCs expressing GFP (*n* = 6, 16 from six independent experiments). **B** *Cxcl12^DsRed+/wt Il15^GFP+/+* double reporter mice show a high co-expression of *Il15* and *Cxcl12* in CD45⁻CD31⁻Lin⁻(TER-119/CD71) stromal cells. Representative flow cytometry plots (left) and relative quantification of IL15-GFP expression among total Cxcl12-DsRed⁺ cells (right); *n* = 4 per group from four independent experiments. **C** Representative flow cytometry plot of CD200⁺CD24⁺ (pre)hypertrophic chondrocytes and CD200⁻CD24⁺ proliferating chondrocytes, pre-gated on CD45⁻Lin⁻ (TER-119/CD71/CD19/CD31), and quantification of GFP-expressing cells in *Il15^GFP+/+* mice (green) or WT controls (grey) (*n* = 5, 10 from 5 independent experiments). **D** Representative

flow cytometry plot of ECs pre-gated on CD31^hiCD144⁺CD45⁻Lin⁻ (Ter119/CD71), showing *Il15^GFP* expression in Sca-1^hi ICAM-1⁻ arteriolar ECs and Sca-1⁺ ICAM-1⁺ sinusoidal ECs of *Il15^GFP+/+* mice (green) or WT controls (grey) (*n* = 6, 16 from 6 independent experiments). **E** Characterization of CD45⁻CD31⁻Lin⁻ (TER-119/CD71) stromal cells in *Il15^GFP+/+* mice: representative histograms (left) of the indicated markers in the GFP⁺ versus GFP⁻ stromal population, and summary dot plot across all experiments showing percent gated as positive (dot size) and the z-score of the median fluorescence intensity for each marker (right). Bar graphs show mean ± SEM with each dot representing a biological replicate. Numbers above the dot plots indicate *P* values from unpaired two-tailed Student's *t* tests. \*\*\**P* < 0.001, \*\*\*\**P* < 0.0001.

osteogenic stromal cell progenitors using *Osx-Cre* mice[7]. Deletion of IL-15 from osteo-lineage stromal cells led to a decrease in NKG2D⁺ rNKP as well as stage A and stage B-C iNK cells but not mNK cells in the BM (Fig. 4A–C). In addition, deletion of IL-15 in *Osx*⁺ stromal cells led to a severe reduction of CD8⁺ $T_{CM}$ and $T_{RM}$ cells and NKT cells in the BM (Fig. 4D–F). This effect was not due to an overall decrease in cell number in the BM nor due to a *Cre* effect, since *Osx-Cre*⁺ mice that were not crossed to *Il15^flox/flox* mice did not show any decline in CD8⁺ $T_{RM}$ or NKT cells in the BM (Fig. S6A, B). *Il15^flox/flox Osx-Cre* mice also showed a decrease in CXCR3⁺ CD8⁺ T cells in the BM, whereas LSKs, CLPs, iILC1 and CD4⁺ $T_{CM}$ cells were not affected (Fig. S6C). A similar reduction of rNKPs and earliest-stage NK cells was also apparent in *Il15ra⁻/⁻* mice, which further progressed with NK cell maturation (Fig. S6D). Deletion of IL-15 in osteo-lineage stromal cells did not affect mNK cells, CD8⁺ $T_{CM}$ and CD4⁺ $T_{CM}$ cells in the blood and spleen, or NKT cells in the spleen (Fig. S6E, F).

We next asked whether differences in the *Osx-Cre* and *Prx1-Cre* phenotype were attributable to off-target effects or differences in the stromal cells themselves. Surprisingly, *Osx-Tomato* fate reporter mice contained a considerable amount of CD45⁺ Tomato^dim cells in digested bone and BM, which showed a high expression of Ly6G and Ly6C (Fig. S7A). We therefore purified various lymphocyte and myeloid populations from the BM of *Il15^flox/flox Osx-Cre* mice and WT littermates to confirm that *Il15* mRNA expression was not reduced in these cell types (Fig. S7B). In contrast to *Prx1-Tomato*⁺ cells, *Osx-Tomato*⁺ cells

were more abundant in digested bone rather than BM (Fig. S7C) and had a higher relative abundance of CD24⁺ cells (Fig. S7D). Strikingly, there was a trend of reduced abundance of Cd1d⁺, Vcam1⁺ and CD200⁺ stromal cells in *Il15^flox/flox Osx-Cre*, but not *Prx1-Cre* mice (Fig. S7E).

Because LepR-expressing stromal cells represent a significant source of IL-15 in the BM, we also analyzed *Il15^flox/flox Lepr-Cre* mice. Our results show that deletion of IL-15 in LepR-expressing stromal cells did not affect the frequency of CD8⁺ $T_{CM}$ cells in 8–13-week-old mice; however, a reduced frequency of CD8⁺ $T_{CM}$ cells was found in 18–26-week-old mice (Fig. S8A). Aging has been shown to lead to an increase in memory CD8⁺ T cells[35,36], but this effect was absent in *Il15^flox/flox Lepr-Cre* mice, despite a slight increase in BM cell numbers (Fig. S8A, B). Another effect that may contribute to this age-dependent effect is that the recombination efficacy of *Lepr-Cre* in the BM has been shown to be low early after birth, with *Lepr-Cre* recombining in only 30% of LepR⁺ cells 14 days postnatal[37]. Although NK cell progenitors, iNK cells, CD8⁺ $T_{RM}$ cells, NKT cells and CD4⁺ $T_{CM}$ cells were not altered, mNK cells and CXCR3⁺ CD8⁺ T cells were decreased in the BM (Fig. S8C, D). No systemic effects were observed, since mNK cells, CD8⁺ $T_{CM}$ and CD4⁺ $T_{CM}$ cells were not altered in the blood and spleen of *Il15^flox/flox Lepr-Cre* mice (Fig. S8E, F). In summary, osteo-lineage stromal cells in the BM support NK cell development (NK cell precursors and iNK cells) as well as the survival of CD8⁺ $T_{CM}$ cells, CD8⁺ $T_{RM}$ cells and NKT cells but not mNK cells and iILCs in the BM. In contrast, LepR⁺ stromal cells had only moderate effects, e.g., partially supporting the survival of mNK cells and CXCR3⁺ CD8⁺ T cells in the BM.

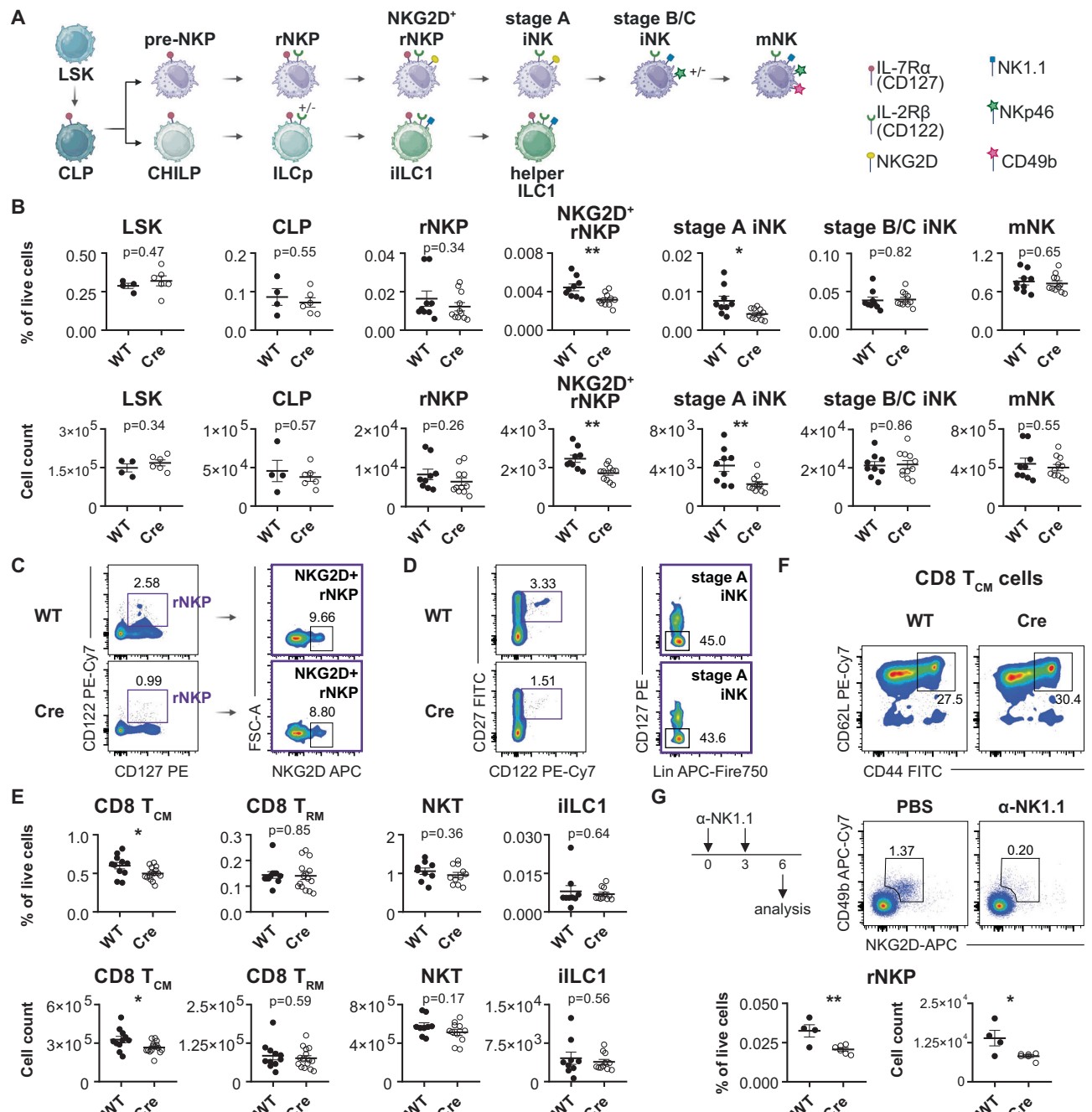

**Fig. 3 | Deletion of Il15 in Prrx1⁺ stromal cells impairs NK cell development and CD8⁺ T_CM cells in the BM. A** Pathway of NK cell and ILC development in the mouse BM. Created in BioRender. Lab Account, D. (2025) https://BioRender.com/wf09zk2. **B** Relative abundance (top) and cell numbers (bottom) of the indicated cell types in the BM of *Il15^flox/flox* and *Il15^flox/flox Prx1-Cre* littermates. Shown are LSK, CLP, NKP and NKP, which co-expressed NKG2D, stage A iNK cells, stage B/C iNK cells and CD49b⁺ mNK cells (*n* = 4–11 from 2–3 independent experiments). **C** Representative flow cytometry plots showing rNKP and NKG2D⁺ rNKP in the BM of an *Il15^flox/flox* and an *Il15^flox/flox Prx1-Cre* littermate. **D** Representative flow cytometry plots showing stage A iNK cells in the BM of *Il15^flox/flox* and *Il15^flox/flox Prx1-Cre* littermate. **E** Relative abundance (top) and cell numbers (bottom) of CD8⁺ T_CM cells, CD69-expressing CD8⁺

T_RM cells, NKT cells and iILC1 in the BM of *Il15^flox/flox* and *Il15^flox/flox Prx1-Cre* littermates (*n* = 11–15 from four independent experiments). **F** Representative flow cytometry plots showing CD8⁺ T_CM cells in the BM of *Il15^flox/flox* and *Il15^flox/flox Prx1-Cre* littermate. **G** Experimental workflow of NK depletion using anti-NK1.1 antibodies and representative flow cytometry plots of NK cell depletion verification in the BM pre-gated on CD45⁺CD3⁻ cells and identified via CD49b and NKG2D (top). Relative quantification and absolute cell numbers of rNKPs in the BM of *Il15^flox/flox* (WT) and *Il15^flox/flox Prx1-Cre* littermates (Cre) after NK cell depletion (bottom); *n* = 4–6 from two independent experiments. All dot plots show mean ± SEM with each dot representing a biological replicate. Numbers above the dot plots indicate *P* values from unpaired two-tailed Student's *t* tests. *\*P* < 0.05, *\*\*P* < 0.01.

## Deletion of IL-15 in ECs impairs NK cells and CD8⁺ T_CM cells in the blood but not in the BM

Next, we deleted IL-15 from ECs, which represented another significant source of IL-15 in the BM. Because *Cdh5-Cre, Tie2-Cre,* and *Flk1-Cre* mice have been known to delete in both ECs and hematopoietic cells[38], we

used inducible *Cdh5-Cre^ERT2* mice provided by Ralf H. Adams[39] to delete *Il15* in ECs only (Fig. 5A). Our results show that conditional deletion in ECs did not affect the frequencies of NKG2D⁺ rNKPs, rNKPs, iNK cells, mNK cells, CD8⁺ T_CM and T_RM cells, iILC1 and NKT cells in the BM (Figs. 5B and S9A, B). However, mNK cells and CD8⁺ T_CM cells, but not

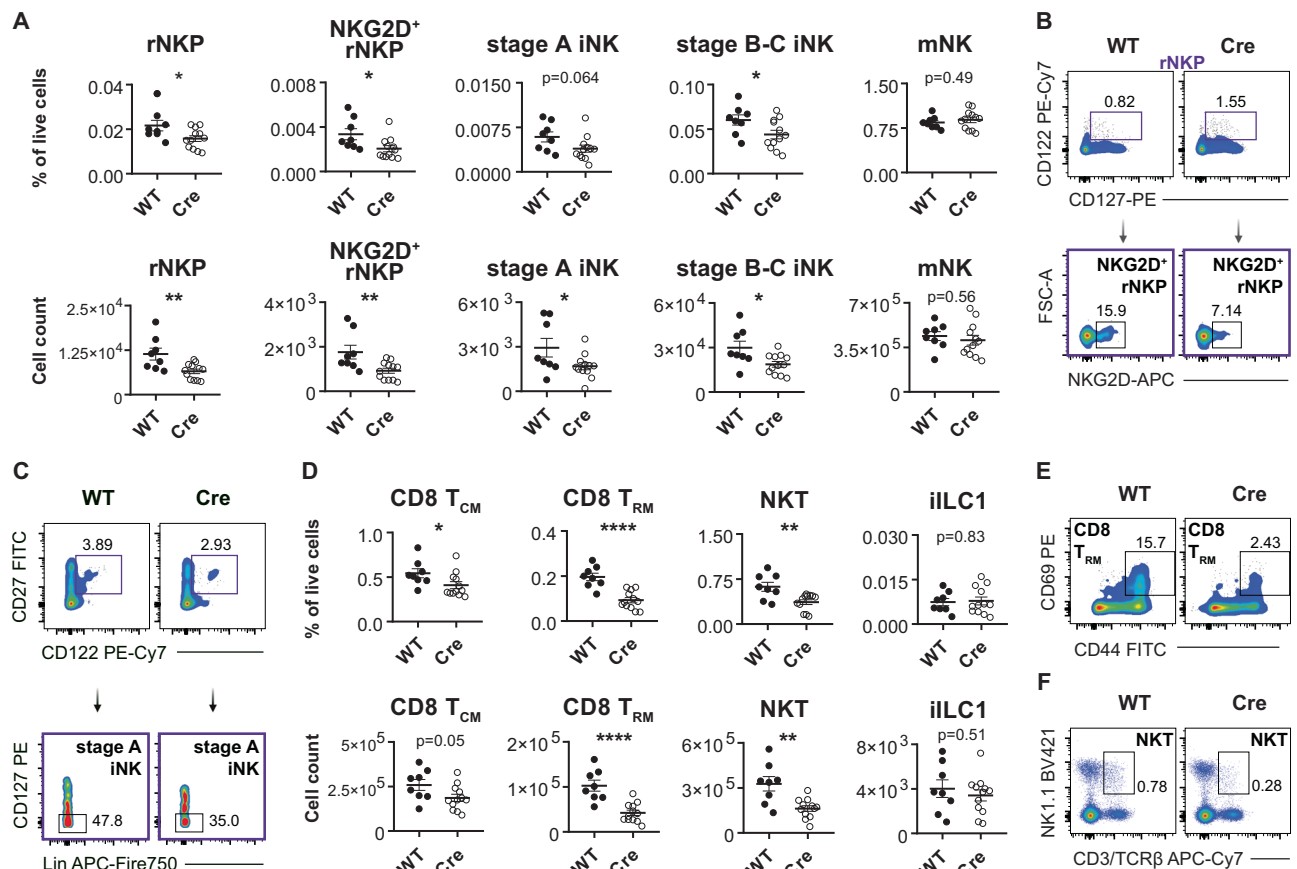

**Fig. 4 | Deletion of Il15 in Osx1⁺ stromal cells impairs NK cell development, NKT and CD8⁺ T$_{CM}$ and T$_{RM}$ cells in the BM. A** Relative abundance (top) and cell numbers (bottom) of the indicated cell types in the BM of *Il15$^{flox/flox}$* and *Il15$^{flox/flox}$ Osx-Cre* littermates. Shown are rNKPs, rNKPs, which co-express NKG2D, stage A iNK cells and CD49b⁺ mNK cells (*n* = 8–12 from five independent experiments). **B** Representative flow cytometry plot of rNKPs from *Il15$^{flox/flox}$ Osx-Cre* and WT littermate. **C** Representative flow cytometry plot of stage A iNK cells from *Il15$^{flox/flox}$ Osx-Cre* and WT littermate. **D** Relative abundance (top) and cell numbers (bottom) of the indicated cell types in the BM of *Il15$^{flox/flox}$* and *Il15$^{flox/flox}$ Osx-Cre* littermates.

Shown are CD8⁺ T$_{CM}$ cells, CD69-expressing CD8⁺ T$_{RM}$ cells, CD3⁺NK1.1⁺ NKT cells and iILC1 cells (*n* = 8–12 from five independent experiments). **E** Representative flow cytometry plot showing CD69-expressing CD8⁺ T$_{RM}$ cells from the BM of *Il15$^{flox/flox}$ Osx-Cre* and WT littermate. **F** Representative flow cytometry plot of CD3⁺NK1.1⁺ NKT cells in the BM of *Il15$^{flox/flox}$ Osx-Cre* and WT littermate. All dot plots show mean ± SEM with each dot representing a biological replicate. Numbers above the dot plots indicate *P* values from unpaired two-tailed Student's *t* tests. \**P* < 0.05, \*\**P* < 0.01, \*\*\*\**P* < 0.0001.

CD4⁺ T$_{CM}$ cell were reduced in the blood (Fig. 5C–E). In the spleen, CD49b⁺ mNK cells but not CD8⁺ T$_{CM}$ and CD4⁺ T$_{CM}$ cells were decreased (Fig. 5F). IL-15 deletion was confirmed on the genetic level after tamoxifen induction (Fig. S9C) and by qPCR in sorted splenic endothelial cells from WT and *Cdh5-Cre$^{ERT2}$* mice (Fig. S9D). Overall IL-15 levels were not significantly reduced in the BM of *Cdh5-Cre$^{ERT2}$* mice, or any of the other tested stromal cKO lines (Fig. S9E). Together, these results indicate that IL-15-expressing ECs support circulating and splenic mNK cells as well as circulating CD8⁺ T$_{CM}$ cells, but do not affect IL-15-dependent immune cell lineages in the BM.

Next, we performed unsupervised subclustering of MSCs from our *Il15$^{GFP}$* scRNA-seq dataset (Fig. S10A, B) in order to decipher receptor-ligand interactions between MSC subtypes and immune cell subsets, in particular NK cell precursors, NKT cells and CD8⁺ T$_{RM}$ cells. We selected a set of nine markers known to play a key role in NK cell precursors[40,41] and analyzed the frequency and level of expression of their ligands in 7 MSC subsets. While *Cxcl12* was expressed in all MSC subsets, NKG2D ligands (*Ulbp1, Raet1d, Raet1e, H60b*), *Il15ra* and *Il18* were predominantly expressed in chondrocytes (Fig. S10C). Querying the public scRNA-seq dataset, we found that 4 MSC subsets, including MSC_osteo and MSC_chondro expressed the non-classical MHC molecule CD1d (Fig. S10D), which were enriched in *Il15⁺* MSCs (Fig. S10E). Flow cytometric analysis revealed considerable CD1d and

*Il15$^{GFP}$* co-expression in stromal cells isolated from BM and bone of *Il15$^{GFP}$* mice (Fig. S10F). Next, we selected a set of 8 markers known to play a key role in CD8⁺ T$_{RM}$ cells[42] and analyzed the frequency and level of expression of their ligands in the 7 MSC subsets. Osteoblasts and chondrocytes showed a high expression of *Tgfb1, Tgfb2, Lgals1, Myl12a* and *Spp1* (Fig. S10G). The coordinated expression of ligands and receptors on MSCs and immune cell subsets provides additional insight into how osteo-lineage MSCs promote the interaction with NK cell precursors, NKT cells and CD8⁺ T$_{RM}$ cells.

**Heterogeneity of IL-15-expressing stromal cells in the human BM**
To explore the frequency and heterogeneity of *IL15-expressing* stromal cells in the human BM, we analyzed 28.183 cells obtained from human BM from 12 individuals that were recently published by Bandyopadhyay and colleagues[43]. The UMAP projection generated from human BM MSC and EC subtypes identified 10 stromal cell clusters, including *THY1⁺* MSCs and Adipo-MSCs, both highly expressing *LEPR* and *CXCL12*, apolipoprotein D (*APOD⁺*) MSCs, arterial ECs (AEC), sinusoidal ECs (SEC) and vascular smooth muscle cells (VSMC) (Fig. 6A, B). Similar to mouse, human *IL15⁺* BM stromal cells were highly abundant in ECs and in *LEPR*-expressing *THY1⁺* MSCs and Adipo-MSCs, with sinusoidal ECs having the highest proportion (~40%) of *IL15⁺* cells within an individual stromal cell type (Fig. 6C, D).

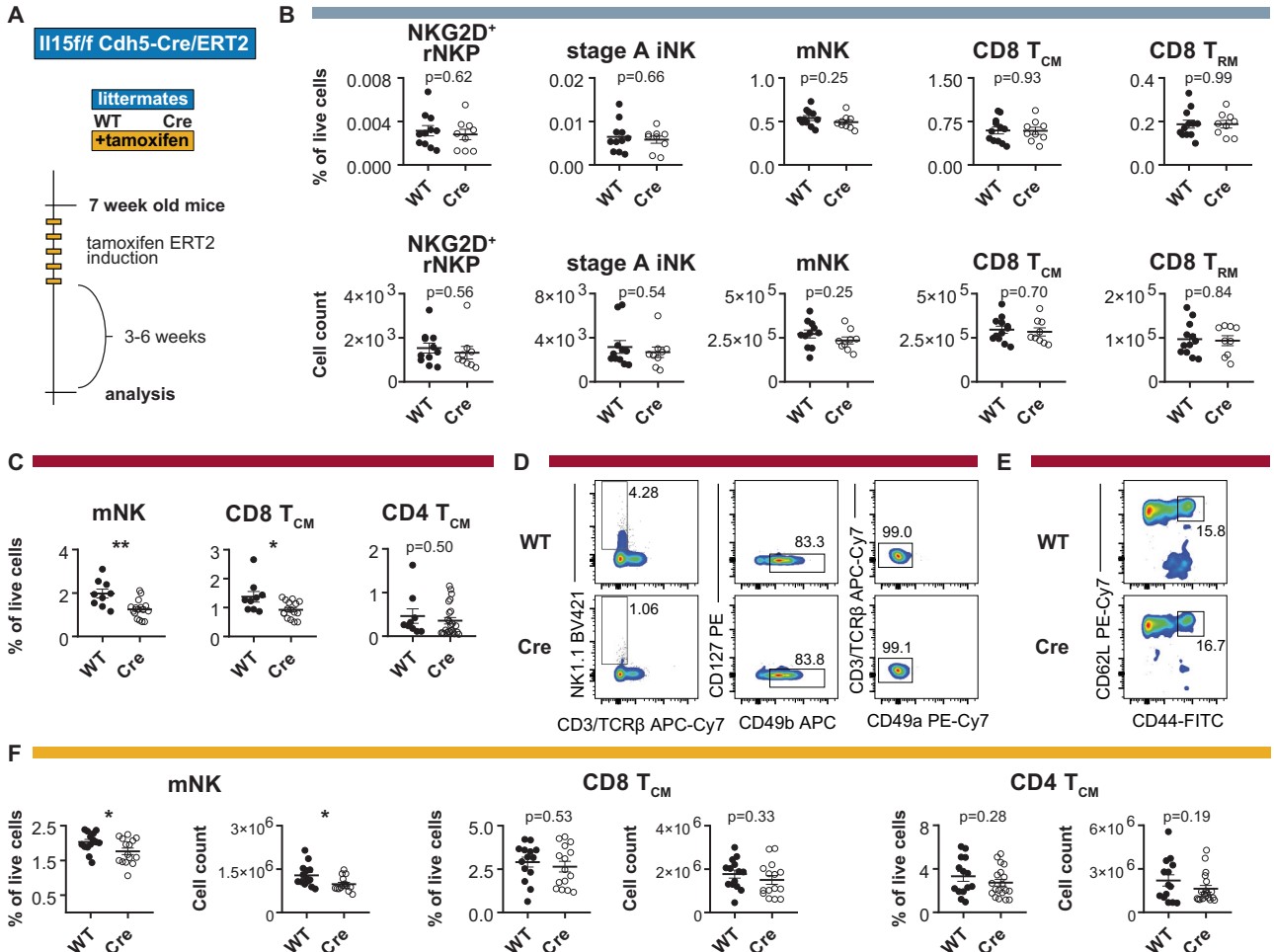

**Fig. 5 | Deletion of Il15 in ECs impairs NK cells and CD8⁺ T_CM in the blood but not in the BM. A** Experimental workflow for tamoxifen-mediated *Cre* induction for endothelial *Il15* deletion using adult *Cdh5-Cre^ERT2* mice. **B** Relative abundance (top) and cell numbers (bottom) of the indicated cell types in the BM (grey bar) of tamoxifen-treated *Il15^flox/flox* and *Il15^flox/flox Cdh5-Cre^ERT2* littermates. Shown are NKG2D⁺ rNKPs, stage A iNK cells, CD49b⁺ mNK cells, CD8⁺ T_CM cells and CD69-expressing CD8⁺ T_RM cells (*n* = 9–12 from three independent experiments). **C** Abundance of CD49b⁺ mNK cells, CD8⁺ T_CM cells and CD4⁺ T_CM cells in the blood (red bar) of tamoxifen-treated *Il15^flox/flox* and *Il15^flox/flox Cdh5-Cre^ERT2* littermates (*n* = 9–16 from four independent experiments). **D** Representative flow cytometry

plots showing NK1.1⁺CD3⁻CD49b⁺CD49a⁻ NK cells in the blood of tamoxifen-treated *Il15^flox/flox Cdh5-Cre^ERT2* and WT littermate. **E** Representative flow cytometry plots showing CD44⁺CD62L⁺ CD8⁺ T_CM cells in the blood of a tamoxifen-treated *Il15^flox/flox Cdh5-Cre^ERT2* and WT littermate. **F** Abundance of CD49b⁺ NK cells, CD8⁺ T_CM cells and CD4⁺ T_CM cells in the spleen (yellow bar) of tamoxifen-treated *Il15^flox/flox* and *Il15^flox/flox Cdh5-Cre^ERT2* littermates (*n* = 14–15 from four independent experiments). All dot plots show mean ± SEM with each dot representing a biological replicate. Numbers above the dot plots indicate *P* values from unpaired two-tailed Student's *t* tests. \**P* < 0.05, \*\**P* < 0.01.

## Discussion

IL-15 plays a key role in NK cell differentiation as well as in the survival of mature NK cells, memory CD8⁺ T cells and NKT cells[12,14–18]. Although some studies suggest a role for non-hematopoietic cells as a source of IL-15[19–22,44], the effects of IL-15-expressing stromal cell subtypes on different immune cell lineages in vivo have not been systematically analyzed.

The low protein expression and femtomolar activity of IL-15 impede typical expression analyses[45] and have since led to the generation of fluorescence reporter mouse models. These include *Il15^EmGFP* BAC transgenic mice[31], and an *Il15^CFP* knockin/knockout model[20], which results in a concomitant knockout of IL-15, as their reporter construct targets the coding region of IL-15 itself. Furthermore, no *Il15^CFP* signal could be detected in sinusoidal ECs, osteoblasts and adipocytes[20,46]. We therefore generated an *Il15-IRES-EGFP* mouse model to ensure concomitant IL-15 and GFP translation while maintaining physiological expression of IL-15. Since the abundance of *Il15*⁺ stromal cells in the BM is low and scRNA-seq is biased toward underestimating weakly expressed genes[47], we generated our scRNA-seq dataset from sorted

*Il15^GFP+* stromal cells. Consistent with the work from others[20,46], CXCL12-abundant reticular (CAR) cells, which expressed LepR and VCAM-1, were the most prominent IL-15-producing stromal cells in the BM. By assessing GFP expression via flow cytometry, 70–80% of LepR⁺ VCAM-1⁺ MSCs produced IL-15 compared to 23% of MSC_stem (published scRNA-seq dataset). ECs, in particular sinusoidal ECs as well as osteo-lineage MSCs (MSC_osteo, chondrocytes), constituted additional significant sources of IL-15 in the BM. Among chondrocytes, only CD200⁺ hypertrophic chondrocytes[48,49], which are anatomically closer to the marrow than proliferating chondrocytes, expressed *Il15^GFP*. Taken together, our *Il15^GFP+* scRNA-seq and flow cytometry analyses reveal distinct IL-15 expression among BM MSC subtypes that may be of physiological relevance.

In order to identify the functional role of stromal cell-derived IL-15, we deleted the cytokine from various non-hematopoietic subsets via Cre-loxP recombination. To avoid overlooking moderate phenotypes that are lost due to compensatory effects, we decided to compare homozygous conditional *Il15* knockout mice (*Il15^flox/flox Cre*⁺ and *Il15^flox/flox* littermates), in contrast to previous studies that used

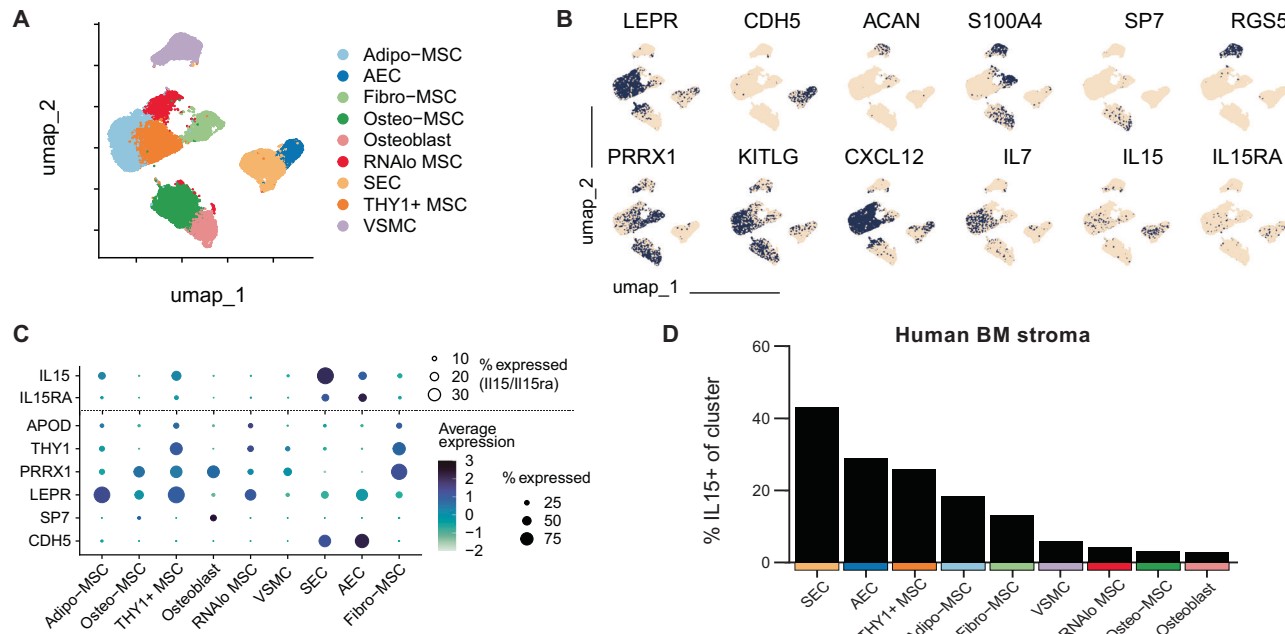

**Fig. 6 | Landscape of Il15-expressing stromal cells in the human BM. A** UMAP projection of 28.183 human BM stromal and endothelial cells obtained from Bandyopadhyay et al.[43]. **B** Feature plots showing key marker genes expressed by the different clusters. **C** Dot plot showing selected differential gene expression by stromal cell clusters identified in the BM. **D** Relative abundance of *Il15*-expressing BM stromal cells on a cluster basis.

heterozygous knockouts as a control group[14,46]. Using *Osx-Cre* mice, which mediate efficient recombination in osteoblast progenitors, osteoblasts, osteocytes and prehypertrophic chondrocytes[50,51], we observed distinct effects on NK cell development (NKG2D⁺ rNKP, stage A-C iNK cells), but not mature NK cells (NK1.1⁺CD49b⁺). A fate mapping study suggested that *Osx-Cre* may also target BM adipocytes[51]. However, a previous study showed that conditional deletion of *Il15* in adipocytes using *Adipoq-Cre* mice did not affect CD49b⁺ or CD49b⁻ NK1.1⁺ NK cells in the BM[21]. Our results also show that conditional deletion of *Il15* in LepR⁺ MSCs or ECs did not affect rNKP and iNK cells in the BM. Although *Prx1-Cre* mice are deleted in all mesenchymal stromal cells[7,34], differences in *Cre*-mediated recombination efficacy depending on the tissue and cell type have been reported. For example, *Prrx1*⁺ cells are able to differentiate into osteoblasts and chondrocytes, however, they only account for 47% of Col1α1⁺ osteoblasts and 6% of Col2⁺ chondrocytes[52]. Accordingly, *Sp7* was frequently and highly expressed in MSC_osteo and chondrocytes in the scRNA-seq dataset, and *Osx-Cre* efficiently targets Sca-1⁻ cells[7] that display a Sca-1⁻ CD51⁺ osteo-lineage marker profile as described previously[53]. This may explain the less pronounced effects observed in *Prx1-Cre* compared to *Osx-Cre* mice, in particular for stage B-C iNK cells, CD8 T_RM and NKT cells. Yet, upon depletion of NK cells using an NK1.1 antibody, *Prx1-Cre* mice couldn't cope with the heightened demand of rNKP generation, and the frequency of CD27⁻CD11b⁺ as well as KLRG1⁺ mature NK cells was decreased.

Previous studies have shown that CXCL12⁺ IL-7⁺ BM MSCs promote HSCs and multipotent progenitors (MPP) in the BM whereas CXCL12⁺ Col1α1⁺ osteoblasts support CLPs and lymphoid-primed MPPs[2,34]. Our results suggest that osteo-lineage MSCs represent a physiologically relevant source of IL-15 for NK cell lineage specification and development in the BM as they promote NKG2D⁺ rNKP and IL-7Rα⁻IL-2Rβ⁺ stage A-C iNK cells. This may further indicate that CLPs, lymphoid-primed MPPs, NKG2D⁺ rNKP and IL-7Rα⁻IL-2Rβ⁺ stage A-C iNK cells are supported by the same osteo-lineage MSC niche in the BM. Data from another study show that conditional deletion of IL-15 from hematopoietic and endothelial cells using *Vav1-Cre* mice partially reduced stage B/C NK1.1⁺ iNK cells but not rNKPs, but this effect was

independent of macrophage- or DC-derived IL-15[46]. Notably, rNKP, NKG2D⁺ rNKP and stage A iNK cells showed a similar reduction in *Il15*^flox/flox *Osx-Cre* and *Il15ra*⁻/⁻ mice, indicating that stromal cells but not hematopoietic cells are the major source of IL-15 to promote NK cell precursors in the BM.

In contrast to other organs[21,54], parenchymal IL-15 appears to play a minor role for mature NK cell survival in the BM, as also suggested by two reports, including a BM chimera study[21,46]. While combined deletion of IL-15Rα from macrophages and DC did not affect NK cell numbers in the BM[14], conditional deletion of IL-15 from hematopoietic and endothelial cells using *Vav1-Cre* mice entirely depleted mNK cells in the BM[46]. Yet, iNKT cells were not affected in *Vav1-Cre* mice[46], indicating distinct survival niches for mNK and iNKT in the BM. Indeed, our results showed that *Osx*⁺ stromal cells supported NKT cell survival in the BM but not in the spleen.

While memory CD4⁺ T cells reside close to IL-7⁺ stromal cells in the BM[55], the survival of BM memory CD8⁺ T cells in the BM has been shown to be in part dependent on macrophage- and DC-derived IL-15, although this effect accounted for only half of the loss of memory CD8⁺ T cells compared to *Il15ra*⁻/⁻ mice[14]. Based on these and other results[44], we speculated that stromal cells or ECs might represent an additional niche for memory CD8⁺ T cell subsets in the BM. Indeed, we observed a decrease of CD8⁺ T_CM cells in the BM of *Osx-Cre*, *Prx1-Cre* and *Lepr-Cre* mice, and a pronounced reduction of CD69-expressing CD8⁺ T_RM cells in osteo-lineage stromal cells. CD69-expressing CD8⁺ T_RM cells are mostly tissue-resident and, unlike CD8⁺ T_CM cells, are not exchanged with the circulating CD8⁺ T cell pool[56,57]. This may explain the profound decrease of CD8⁺ T_RM cells in *Osx-Cre* mice and may also indicate that CD8⁺ T_RM cells are limited in their ability to translocate within the BM and utilize IL-15 from other cellular sources. CXCR3 has been shown to be expressed in skin-resident memory T cells[58], and the CXCR3 ligands CXCL9 and CXCL10 are mostly expressed in MSC_stem and sinusoidal ECs but not MSC_osteo, chondrocytes and fibroblasts (scRNA-seq dataset). This may explain why CXCR3⁺ CD8⁺ T cells are in part dependent on IL-15 from *Lepr*⁺ and *Prrx1*⁺ stromal cells in the BM. CXCR4, the ligand for CXCL12, is not expressed in CD8⁺ T_RM cells[59], which may indicate a role for CXCL12⁻ IL-15⁺ osteo-lineage MSCs rather

than CXCL12[+] LepR[+] MSCs in supporting CD8[+] T$_{RM}$ cell survival in the BM. Further circumstantial evidence as to why IL-15[+] osteo-lineage MSCs maintain CD69-expressing CD8[+] T$_{RM}$ cells in the BM might be the preferential expression of the CD69 ligand galectin-1[60] by MSC_osteo and fibroblasts[61] and the CD69 ligands S100a8 and S100a9 by chondrocytes[62].

ECs have been shown to play a key role in the survival of HSCs and early lymphoid progenitors in the BM via CXCL12[6,34]. We show that ECs, in particular ICAM-1[+] sinusoidal ECs, represent a significant source of IL-15 in the BM. Yet, ECs had a lower Il15-GFP expression intensity compared to VCAM-1[+] MSCs. As the primary site of leukocyte trafficking in and out of the BM[63], sinusoidal ECs might also affect IL-15-dependent trafficking, a process which has been described for memory T cells[64]. Notably, specific deletion of IL-15 in ECs using Cdh5-Cre$^{ERT2}$ mice had no effect on any of the tested immune cell subsets in the BM. However, circulating mNK cells and CD8[+] T$_{CM}$ in the peripheral blood were significantly decreased. These results indicate that EC-derived IL-15 did not affect developmental or survival niches of IL-15-dependent immune cell lineages in the BM.

In summary, our data illustrate heterogeneity of IL-15 expression in BM MSCs and demonstrate that conditional deletion of Il15 in different MSC lineages characteristically affects the development and survival of IL-15-dependent immature and mature immune cell subsets in the BM.

## Limitations of this study

Despite high Il15 and Il15$^{GFP}$ expression in LepR[+] VCAM-1[+] MSCs, the effects of IL-15 deletion in Lepr-Cre mice were moderate, in line with a recent study[46]. The Lepr-Cre strain used in this study contains an IRES-NLS-Cre at the endogenous locus and likely expresses Cre only in the canonical LepR-B isoform[65]. The low expression level of LepR observed by flow cytometry may lead to insufficient Cre-mediated recombination in LepR$^{dim}$ MSCs, a problem that has been commonly observed when using Cre lines[66], and that may underestimate the role of IL-15 produced by LepR[+] BM MSCs. Furthermore, Lepr-Cre-mediated deletion has recently been reported to have a delayed onset in the BM[37], which may explain why 18–26 week old but not 8–13 week old Il15$^{flox/flox}$ Lepr-Cre$^{+/wt}$ mice showed a phenotype. Additionally, Ciu et al. reported that Il15 expression markedly increased in VCAM-1[+] BM stroma cells with age[20] and coincides with the age-dependent increase in memory CD8[+] T cells in the BM[35,36]. Indeed, we observed a significant reduction of CD8[+] T$_{CM}$ and mNK cells in the BM of 18–26 week old mice, which might be attributed to the inhibition of an age-related accumulation of these immune cell subsets[67].

## Methods
### Mice
All experiments were approved by the animal ethics committee of the Medical University of Vienna and the Austrian Federal Ministry of Education, Science and Research (GZ 66.009/0407-V/3b/2018 and GZ 66.009/0408-V/3b/2018). Il15$^{GFP}$ homozygous reporter knockin mice were generated by D.H.-B. in the lab of Richard A. Favell (Yale University, USA; MTA #20104) and are available upon request from the Flavell laboratory. Il15$^{flox/flox}$ mice were generated by Nan-Shih Liao[21] (Academia Sinica, Taiwan; MTA #13T-1050130-16M) and purchased from The Jackson Laboratory (Stock No. 034188). Cdh5-Cre$^{ERT2}$ mice were kindly provided by Ralf H. Adams[39] (Max Planck Institute for Molecular Biomedicine, Germany; MTA with CancerTools.org catalogue number 151520). Prx1-Cre[33] (Stock No. 005584), Lepr-Cre[68] (Stock No. 008320), Osx1-GFP-Cre[69] (Stock No. 006361), Ai14-Tomato[70] (Stock No. 007914) and CXCL12-DsRed knockin/knockout mice[34] (Stock No. 022458) and Il15ra$^{-/-}$ (Stock No. 003723) were purchased from The Jackson Laboratory. Conditional knockout lines were generated by interbreeding Il15$^{flox/flox}$ with the respective Cre lines. Age- and sex-matched Cre$^-$ and Cre$^+$ mice (littermates wherever possible) have been used in all experiments. Cre$^+$ only mice (not crossed with floxed strains) have been used as an additional control group to exclude any potential effects mediated by Cre recombinase. Deletion-specific primers of the Il15 locus were included in the genotyping workflow to monitor and avoid the propagation of germline deletions. All primers used for genotyping are indicated in Table S1. For experiments of tamoxifen-inducible expression of Cre, 1 mg of tamoxifen (Sigma-Aldrich, Cat# T5648) was intraperitoneally injected into 7–8-week-old mice for five consecutive days, and mice were analyzed 3–6 weeks after the last day of induction. Animals were housed in individually ventilated cages at a temperature of $22 \pm 2$ °C, humidity of $55 \pm 10\%$, and a 12-hour dark/light cycle. Mice were euthanized by cervical dislocation.

### Generation of Il15-IRES-EGFP knockin mice
Il15-IRES-EGFP mice were generated using a standard cloning strategy, and the animal studies were performed in accordance with the guidelines of the Office of Animal Research Support of Yale University (New Haven, CT). The BAC clone RP23-79L16, consisting of Il15 genomic DNA derived from C57BL/6 mice, was purchased from BACPAC Genomics (Oakland, CA). Homologous arms of the Il15 locus were generated from genomic DNA by PCR. A 2.965 kb fragment containing exons 7 and 8 was used as the 5′ homology region, and a 2.265 kb fragment containing 3′-UTR was used as the 3′ homology region. The targeting construct was generated by cloning the 5′-arm into XhoI sites directly after the translation stop codon (TGA) of the Il15 gene and the 3′-arm into NotI sites of the pEasy-Flox IRES eGFP vector linked to a LoxP-flanked neomycin (Neo) selection marker[71]. The targeting construct was linearized by PvuI cleavage and subsequently electroporated into Bruce4 C57BL/6 embryonic stem (ES) cells. Transfected ES cells were selected in the presence of 300 µg/mL G418 and 1 µM ganciclovir. Drug-resistant ES cell clones were screened for homologous recombination by PCR. To obtain chimeric mice, correctly targeted ES clones were injected into blastocysts, which were then implanted into CD1 pseudopregnant foster mothers. Male chimeras were bred with C57BL/6 to screen for germline-transmitted offspring. Germline-transmitted mice were then bred with Tet-Cre transgenic mice to remove the neomycin gene. Mice bearing the targeted Il15 allele were screened by PCR (see Table S1 for primer sequences).

### Tissue collection and generation of single-cell suspensions
BM cells were isolated from the femur and tibia of 8–16-week-old mice (if not explicitly indicated otherwise). For immune cell panels, BM was thoroughly flushed out of the bones with ice-cold FACS buffer (1× PBS, 2% FBS), centrifuged for 5 min at $400 \times g$ and then incubated in 2 mL ACK lysis buffer (150 mM NH$_4$Cl, 10 mM KHCO$_3$, 0.1 mM Na$_2$EDTA, pH 7.4) for 5 min at RT to lyse erythrocytes. After a second centrifugation step, samples were resuspended in an appropriate volume of FACS buffer and Fc receptors were blocked with TruStain FcX Plus (BioLegend, Cat# 156604) before adding the specific surface antibodies for 30–60 min on ice. Samples were finally washed two times with 2 mL FACS buffer. BM stromal cells were isolated using an enzymatic digestion protocol and handled in tubes pre-coated with FBS overnight at 4 °C to reduce adherence to the plastic surfaces. Briefly, femurs and tibias were dissected from euthanized mice, and soft tissue was carefully removed. BM was flushed with staining buffer (PBS supplemented with 10% FBS and 2 mM EDTA), and the remaining bones were crushed using a mortar and pestle. The pelleted BM and bone fractions were then digested for 30–40 minutes at 37 °C by using collagenase IV at 1 mg/mL (Sigma-Aldrich, Cat# C5138), DNase I at 1 mg/mL (Sigma-Aldrich, Cat# 10104159001) and Dispase at 0.5 mg/mL (Roche, Cat# 04942078001) in Dulbecco's modified eagle medium. The digestion mix was filtered through a 70 µm cell strainer to remove debris and bone fragments and neutralized with staining buffer. The filtered cell suspension was centrifuged at $300 \times g$ for 8 minutes and resuspended in appropriate volume of staining buffer, Fc-blocked, stained, and

washed. To obtain splenocyte single cell suspensions, spleens were gently mashed using the flat end of a plastic 10 mL syringe and filtered through a 70 μm mesh cell strainer with 3 mL FACS buffer. Peripheral blood was obtained by retro-orbital bleeding of isoflurane-anaesthesized mice, ACK-lysed twice for 5 min at RT and subsequently resuspended in FACS buffer. Cell concentrations (BM of two femurs and tibias or whole spleen) were determined by counting trypan blue negative cells on a hemocytometer (Neubauer). In some instances, stromal cells were magnetically pre-enriched using the EasySep Mouse PE Positive Selection Kit (StemCell Technologies, Cat# 17666), e.g., for qPCR of CD73-PE⁺ $Il15^{GFP+}$ and $Il15^{GFP-}$ sorted cells.

### Flow cytometry

Single cell suspensions were acquired on a BD LSR Fortessa X20 flow cytometer (BD Biosciences) using BD FACSDiva software. The following antibodies were obtained from BioLegend: α4β7 (DATK32, Cat# 120608, 1:100), Bcl-2 (BCL/10C4, Cat# 633510, 1:67), CD1d (1B1, Cat# 123527, 1:67), CD3e (145-2C11, Cat# 100355, 500A2, 1:100, Cat# 152308, 1:100), CD4 (RM4-5, Cat# 100536, 1:100), CD8a (53-6.7, Cat# 100732, 1:50, 100742, 1:200), CD11b (M1/70, Cat# 101230, 101206, 1:100), CD11c (N418, Cat# 117352, 1:100), CD19 (1D3/CD19, Cat# 152412, 1:100), CD24 (30-F1, Cat# 138505, 1:100), CD25 (3C7, Cat# 101916, 1:100), CD27 (LG.3A10, Cat# 124207, 1:100), CD31 (MEC13.3, Cat# 102523, 1:100), CD44 (IM7, Cat# 103006, 1:200), CD45 (30-F11, Cat# 103128, 103154, 1:100), CD49a (HM1a, Cat# 142608, 1:100), CD49b (DX5, Cat# 108910, 1:100), CD51 (RMV-7, Cat# 104106, 1:100), CD54 (YN1/1.7.4, Cat# 116120, 1:100), CD62L (MEL-14, Cat# 104417, 1:200), CD69 (H1.2F3, Cat# 104507, 1:67), CD71 (R17217, Cat# 113828, 1:100), CD73 (TY/23, Cat# 117204, 1:100), CD106 (429, Cat# 105712, 1:100), CD117 (2B8, Cat# 105841, 1:67 and ACK2, Cat# 135122, 1:67), CD122 (TM-β1, Cat# 123219, 1:100), CD127 (A7R34, Cat# 135009, 1:40), CD135 (A2F10, Cat# 135314, 1:67), CD144 (BV13, Cat# 138013, 1:67), CD146 (ME-9F1, Cat# 134704, 1:100), CD200 (OX-90, Cat# 565547, 1:100), CXCR3 (CXCR3-173, Cat# 126522, 1:67), F4/80 (BM8, Cat# 123110, 1:67), Gr-1 (Rb6-8c5, Cat# 108423, 1:200), IFN-γ (XMG1.2, Cat# 505810, 1:67), Ki-67 (11F6, Cat# 151227, 1:100), KLRG1 (2F1/KLRG1, Cat# 138412, 1:100), Ly6C (HK1.4, Cat# 128017, 1:200), Ly6G (1A8, Cat# 127605, 1:200), NK1.1 (PK136, Cat# 108732, 1:100), NKG2D (CX5, Cat# 115711, 1:100), Sca-1 (D7, Cat# 108105, 1:67), TCRβ (H57-597, Cat# 109246, 1:67), TER-119 (TER-119, Cat# 116223, 116206, 116233, 1:100), TNF-α (MP6-XT22, Cat# 506303, 1:100). CD8a (53-6.7, Cat# 566410, 1:200) was obtained from BD Biosciences and LepR (polyclonal, Cat# BAF497, 1:67) from R&D Systems. Ghost Dye V510 (Tonbo Biosciences, Cat# SKU 13-0870-T100, 1:200) or Zombie Aqua (BioLegend, Cat# 423101, 1:200) were used as a viability stain. Chondrocytes were gated as described before[48,49]. Nuclear antigens were stained using the eBioscience Foxp3/Transcription Factor Staining Buffer Set according to the manufacturers' instructions (Thermo Fisher Scientific, Cat# 00-5523-00). Data were analyzed using FlowJo V10 (BD Biosciences). Absolute numbers were calculated by cell counting and determining relative amounts using FlowJo software.

### In vitro activation for intracellular cytokine staining

BM cell suspensions were stimulated in vitro with PMA/ionomycin (1x Cell Activation Cocktail, BioLegend, Cat# 423301) in the presence of 5 μg/mL Brefeldin A (BioLegend, Cat# 420601) for 3 hours and then fixed in Fixation Buffer (BioLegend, Cat# 420801) prior to permeabilization using Permeabilization Wash Buffer and intracellular cytokine staining.

### Elisa

ACK-lysed BM extracts were homogenized in non-denaturing lysis buffer (20 mM Tris pH 7.5, 150 mM NaCl, 1 mM EDTA, 1% Triton X-100, 1% Halt Proteinase Inhibitor from Thermo Fisher Scientific) for 30 min on ice and total protein concentration was measured using the Pierce

bicinchoninic assay (BCA) kit (Thermo Fisher Scientific, Cat# 23227). IL-15 protein concentrations in these samples were determined using the IL-15 Duo Set ELISA (Cat# DY447-05) from R&D Systems. Sample concentrations were interpolated by using a 5-parameter logistic (5PL) curve in GraphPad Prism, and total amounts were normalized to total protein concentration as determined with BCA.

### NK cell depletion

Selective depletion of NK cells was accomplished by intraperitoneally injecting two doses of 25 μg purified anti-NK1.1 antibody (BioLegend, clone PK136, Cat# 108702) or PBS 3 days apart. Organs were harvested and analyzed 4 days after the last injection.

### qPCR

RNA was extracted from whole bone single cell suspensions or sorted cells (sorted directly into RLT Plus lysis buffer supplemented with 1% β-mercaptoethanol) using the RNeasy Mini or RNeasy Micro Plus kits from Qiagen according to the manufacturer's instructions. cDNA was synthesized with the RevertAid First Strand cDNA synthesis kit (Thermo Fisher Scientific), using the supplied oligo(dT)₁₈ primers. Quantitative reverse transcription PCR (qRT-PCR) was performed on a CFX96 Touch Real-Time PCR Detection System (BioRad) using Maxima SYBR Green master mix including ROX reference dye (Thermo Fisher Scientific). Sequence-specific oligonucleotide primers were synthesized by Sigma Aldrich. Relative expression values were normalized to mouse *beta-actin* and *Gapdh* using the comparative threshold cycle method ($2^{-\Delta Ct}$, or $2^{-\Delta\Delta Ct}$ for fold changes between samples, e.g., $Il15^-$ versus $Il15^+$ MSCs). Primers used are described in Table S2.

### Single-cell RNA sequencing of primary stromal cells

**Isolation and purification of stromal cells.** A single cell suspension of stromal cell-containing BM and bone was generated by enzymatic digestion as described above. Cell suspensions were incubated with TruStain FcX Plus (BioLegend) for 5 min and then stained on ice for 60 min. Viable *Il15*-expressing stromal and endothelial cells were stained and sorted on a FACSMelody cell sorter (BD Biosciences) using Zombie Aqua⁻ CD45⁻ TER-119⁻ and CD31⁻ $Il15^{GFP+}$ (MSCs) and CD31⁺ $Il15^{GFP+}$ (endothelial cells). Cells were sorted into cooled collection tubes supplied with 50 μl collection medium (PBS with 0.08% BSA). The obtained data are from two independent experiment days and 3 sequencing runs and contain BM- and bone-derived cells from a total of seven $Il15^{GFP}$ mice (male and female, 8–12 weeks old).

**Library preparation and sequencing.** The sorted cells were immediately processed for scRNA-seq on a Chromium Controller using the Single-Cell 3' Gel-Bead Kit v3.1 and library kit (10X Genomics, Pleasanton, CA) following the manufacturer's instructions. Libraries were sequenced on an Illumina NovaSeq 6000 platform using the 50 bp paired-end setting. Sequencing data were processed and aligned to the mouse genome version mm10 using the Cell Ranger Single cell software suite (version 7.1.0, 10x Genomics) using intron mode and including a custom reference genome including the *Il15-IRES-EGFP* construct.

**Data analysis.** Analysis of $Il15^{GFP+}$ BM stromal cells and public BM stromal cell datasets was performed using the R Bioconductor package Seurat (Version 5.0.1). Quality control included the exclusion of cells with a count of unique genes under 300 and over 6000, and a fraction of mitochondrial genes <0.1. Furthermore, contaminating hematopoietic cells were excluded on a cluster basis by omitting hematopoietic clusters expressing *Ptprc* or *Tfrc* and additionally by excluding cells with an expression value > 0 of the erythro-lineage marker *Gypa*. Individual datasets were integrated using Harmony. UMAP embedding and Louvain clustering were employed at a resolution of 0.2 and 30 PCA dimensions. The markers shown in Figs. 1B, G and 6B were queried

as Seurat Feature Plots with a maximum cutoff of the 70th percentile for better visualization of weakly expressed transcripts. We then used the information from the public dataset for the analysis of our *Il15*<sup>GFP+</sup>-enriched stromal cells by using the FindTransferAnchors function implemented in Seurat, which enables the identification of shared cellular subpopulations between datasets. Specifically, we utilized the public dataset as a reference to align the new dataset onto a common feature space, and used TransferData, to predict cluster labels based on their similarity to reference cells in the public dataset. The re-analyzed, integrated and quality-controlled public dataset includes a total of 68.616 cells from various biological samples under steady state conditions (BM and bone from 3 to 16-week-old C57BL/6 wild type mice or *Col2.3-Cre, Lepr-Cre, VEC-Cre* fate reporter lines on a C57BL/6 background).

For directly comparing IL-15 expression in *Il15*<sup>GFP+</sup> sorted versus total stromal cells (Fig. S1E–G), gene symbols between datasets were harmonized using the HGNChelper R package before merging and integrating the individual Seurat objects using Harmony. Non-stromal cells were excluded after merging of these datasets, first on a cluster basis by omitting hematopoietic clusters expressing *Ptprc* or *Tfrc*, and additionally by excluding cells that had an expression >0 of *Gypa, Ppbp, Cd79a* and *Cd79b*, or *Elane* (leading to differing cell numbers for the individual datasets in the public vs *Il15*<sup>GFP</sup> vs meta-dataset).

In order to investigate IL-15 expression in human BM stroma, we re-analyzed a publicly available scRNA-seq dataset[43], subsetted the Seurat object on stromal clusters including endothelial cells, removed clusters containing *PTPRC* and *CD79B*-expressing cells, and normalized and scaled the dataset using the default Seurat 5.1.0 settings. Data were then integrated on a patient basis using Harmony. UMAP embedding and Louvain clustering were employed at a resolution of 0.4 and 15 PCA dimensions.

## Statistics and reproducibility

Data are presented as the mean ± SEM. A *p* value below 0.05 was considered significant. Two-tailed student's *t* tests were calculated for the comparison of two groups. One-way analysis of variance (ANOVA) with Tukey's or two-way ANOVA with Sidak's multiple comparison post-hoc tests was used for testing multiple groups. Statistical analysis was performed using GraphPad Prism 10.2.3. No statistical method was used to predetermine sample size. PCR genotyping determined group allocation, mice were processed in a random order, and multiple investigators were involved in data generation and analysis. Data from each genotype were from at least two independent experiments and at least three different litters.

## Reporting summary

Further information on research design is available in the Nature Portfolio Reporting Summary linked to this article.

## Data availability

The single-cell RNA sequencing data generated in this study have been deposited in the Gene Expression Omnibus under accession code GSE273212. All other data supporting the findings of the present study are available from the corresponding author upon reasonable request. The publicly available scRNA-seq datasets re-analyzed in Fig. 1 were obtained from ref. 29 (GSE156635; samples GSM4735393, GSM4735394, GSM4735395, GSM4735396), ref. 30 (GSE122467, samples GSM3466899, GSM3466900, GSM3466901, GSM3937216, GSM3937217), ref. 9 (GSE108892, samples GSM2915575, GSM2915576, GSM2915577, GSM2915578, GSM2915579, GSM3494769, GSM3494771) and ref. 10 (GSE128423; samples GSM3674224_std1, GSM3674225_std2, GSM3674226_std3, GSM3674227_std4, GSM3674228_std5, GSM3674229_std6, GSM3674239_b1, GSM3674240_b2, GSM3674241_b3, GSM3674242_b4, GSM3674243_bm1, GSM3674244_bm2, GSM3674245_bm3, GSM3674246_bm4). The publicly available scRNA-seq dataset

re-analyzed in Fig. 6 was obtained from ref. 43 (GSE253355). Source data are provided with this paper.

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

## Acknowledgements

We would like to thank Nan-Shih Liao (Academia Sinica, Taipei, Taiwan) for sharing *Il15^flox/flox* mice[21] and Ralf H. Adams (Max Planck Institute for Molecular Biomedicine, Münster, Germany) for sharing *Cdh5-Cre^ERT2* mice[39]. We would like to thank Judith Stein, Linda Evangelisti and Cynthia Hughes (Yale University, New Haven, USA) for their help in generating *Il15^GFP* mice, Valerie Plajer, Melanie Lietzenmayer, Marina Schernthanner, Johannes Reisecker and the Vienna Biomedical Sequencing Facility (Benjamen White; Head: Christoph Bock) for technical assistance. We would like to thank Jonathan Alderman, Robert Eferl, Wolfgang P. Weninger, Maria Sibilia and the members of the Herndler-Brandstetter lab for their support and advice. C.S. was supported by an APART-MINT postdoctoral fellowship of the Austrian Academy of Sciences. R.B. and M. Frank were supported by a DOC fellowship of the Austrian Academy of Sciences. This research was funded in whole or in part by the Austrian Science Fund (FWF) [10.55776/F61] to M. Farlik, and [10.55776/P33340] and [10.55776/P36995] to D.H.-B. The work was supported by the Vienna Science and Technology Fund (WWTF) [10.47379/LS20042] to D.H.-B. and M. Farlik.

## Author contributions

C.S., R.B., R.A.F., and D.H.-B. designed the experiments. C.S., R.B., A.S.-B., S.F., E.P., N.B., M. Farlik, and D.H.-B. performed the experiments. C.S., R.B., and D.H.-B analyzed the data. C.S. performed the analysis of the scRNA-seq data. R.B. and M. Frank helped with the computational analysis. C.S. and D.H.-B. wrote the manuscript. R.A.F. supervised the generation of *Il15^GFP* mice. D.H.-B. supervised the work. All authors discussed the data and commented on the manuscript.

## Competing interests

The authors declare no competing interests.
