## [Transparent Peer Review file · Nature Communications]

Heterogeneity of IL-15-expressing mesenchymal stromal cells controls natural killer cell development and immune cell homeostasis

Corresponding Author: Dr Dietmar Herndler-Brandstetter

Version 0:

Reviewer comments:

Reviewer #1

(Remarks to the Author)

It has been previously shown that conditional deletion of IL-15 in *Lepr*⁺ cell-derived stromal cells or *Prx1*⁺ cell-derived stromal cells, including *Osterix*⁺ stromal cells, *Lepr*⁺ stromal cells, and osteoblasts, did not affect the numbers of NK cell precursors and mature NK cells but that conditional deletion of IL-15 in hematopoietic cells markedly reduced the numbers of NK cell precursors and mature NK cells (ref. 46; Abe et al., Cell reports 2023). The authors generated novel IL-15 knockin reporter mice, in which physiological expression of IL-15 was maintained and show IL-15 expression levels in various non-hematopoietic cell populations in the bone marrow. In addition, they show that conditional deletion of IL-15 in *Osterix*⁺ cell-derived stromal cells decreased slightly NK cell precursors, memory CD8⁺ T cells and NKT cells but not mature NK cells and that conditional deletion of IL-15 in *Lepr*⁺ cell-derived stromal cells decreased slightly memory CD8⁺ T cells and mature NK cells at the age of 18-26 weeks but not 8-13 weeks. This is an important, in-depth performed, and well-written study; however, the paper would fall a bit short of providing an advance over the previous work. Most novel aspect of the paper is the data on memory CD8⁺ T cells; however, differences seen would be small and not very impressive.

Reviewer #2

(Remarks to the Author)

In this manuscript, Stecher and colleagues investigate the cellular sources of IL-15 in the bone marrow (BM) and the functional contribution of different IL-15 producing cell types to NK cell development and CD8 memory T cell (CD8 TCM) homeostasis. Through integration of previously published sc-RNAseq datasets of the BM stromal compartment, the authors report expression of *Il15* transcripts primarily in various types of endothelial and mesenchymal stromal cell (MSC) populations. They generate a new reporter model in which expression of IL15-GFP is detected mostly in non-hematopoietic stromal cells, including *LepR*⁺ MSCs (also known as *Cxcl12*-abundant reticular cells-CARc), sinusoidal endothelial cells (SECs), chondrocytes and osteoprogenitor cells. To determine the functional role of these cell types in IL15-dependent regulation of hematopoiesis, the authors generate cell-type specific IL-15-deficient mouse models by crossing *IL15^{fl/fl}* mice to different Cre lines, including *Prx-1-Cre*, *LepR-Cre*, *Osx-Cre* and *Cdh5-Cre* strains. Analyses of these mouse lines lead the authors to the general conclusion that IL-15 derived from MSC subsets primarily controls both NK cell development and CD8 TCM maintenance in the BM, while only having a minor or negligible impact on the numbers of circulating cells. They also find that EC-specific deletion of IL-15 causes a decrease of mature NK cells and CD8 Tcm cells in the blood, while not having any effects on BM populations. Finally, inspection of published datasets of human BM stroma suggests that the profile of expression of IL-15 in MSC and EC subsets is largely conserved between murine and human marrow.

Dissecting the specific contribution of stromal subsets to the regulation of distinct developmental stages of hematopoiesis is of high relevance. The organization of BM into functional, spatially restricted niches suggests that secretion of the same chemokine by different cell types may serve to control different developmental processes, which maybe spatially segregated. This notion has been experimentally confirmed in the case of *Cxcl12* and *Scf*. Yet, whether IL-15 secretion by different subsets in specific niches plays different roles is unknown. Nonetheless, in its current form, this study falls short in clarifying this crucial question. Detailed comments are provided below:

- Although the selection of murine models targeting different stromal compartments seems suitable a priori, the results obtained in the MSC-specific strains are hard to interpret and integrate into a coherent model. A thorough characterization of the mouse strains in terms of which populations and tissue regions are targeted is lacking in the manuscript. As a prime example, the Prx1 promoter has been previously shown to target all mesenchymal subsets with high efficiency, which also include the entirety of osteolineage cells that are targeted by the Osx-Cre mouse model, as well as all MSC progenitor cell populations. Based on this expected overlap, it would be anticipated that the results of deleting IL15 in the Prx-1Cre model would, at least phenocopy, if not exceed in magnitude, those observed in the Osx-1Cre line. Yet, the authors observe that the impact on NK cell development is more pronounced in Osx-Cre mice than in Prx1-Cre mice. The authors point out that osteoblasts and chondrocytes are only partially targeted in Prx1-Cre mice. However, this point is incorrect, as they refer to a report in which the targeting efficiency was analyzed in Prx1-CreERT2, but not Prx-1-Cre mice, while robust evidence suggests that the use of Prx1-Cre mice very efficiently targets all MSC derivatives in hind limbs. To explain the discrepancies between both models, the authors should thoroughly evaluate the targeting efficiency in MSC subsets of all the mouse lines employed by flow cytometry (by crossing to a ROSA26-tdTom mouse line) and results should be validated via imaging, which would also provide some spatial insight that could reveal the specific niches in which the cytokine is produced (and depleted in Cre lines) and therefore guide the interpretation of the results here obtained. This in-depth characterization should also comprise stromal cells, to make sure that Il15 deletion is not affecting numbers and frequencies of relevant stromal subsets. Of major importance, it would be key to understand how targeting of the different populations affects the overall levels of IL-15 and IL-15RA in BM supernatants, which may be measured using ELISA techniques.
- A previous publication by Abe et al. (2023 Cell Reports 42, 113127) had already reported a lack of significant defects in NK cell development in LepR-Cre Il15fl/fl and Prx-1Cre Il-15fl/fl mice. How do the authors explain the discrepancies with these reports (especially in the case of Prx-1Cre mice) ? In this same report, NK cells and progenitors are found either scattered or in clusters in BM tissues. It would be relevant to understand whether the defects found in Osx-Cre mice apply to both cellular distributions.
- The effects of selectively depleting Il15 in MSC subsets, on NK cell development are relatively minor and selectively apply to progenitor stages while sparing mature NK cells. How do the authors envision that developmental defects are compensated in mature subsets?
- Most importantly, given the magnitude of the effects observed both in NK as well as CD8TCM cells it would be crucial to test the functional consequences of these minor reductions in a pathological setting when increased numbers of these cellular subsets need to be mobilized.
- The expression patterns of the new IL-15 reporter mouse model do not fully recapitulate those observed in previously generated mouse reporter lines. More specifically, no expression of IL-15 in ECs and osteoblasts was found in these previous reports. Also, 90% of LepR+ cells were shown to express the cytokine compared to the 65% observed here. Given these discrepancies, and considering that a large fraction of sorted GFP+ cells appear not to express Il15 transcripts in the sc-RNAseq data (Figure 1F), the authors should carefully assess using qRT-PCR, to what extent GFP expression reflects that of IL-15 in their mouse model in all of the cell types studied.
- The mouse models employed to target MSCs constitutively express the Cre transgene, and therefore expression of Il15 is targeted throughout development. Therefore, some of the effects observed could be derived from the impact of IL-15 deletion in NK progenitors at these early stages. This point should be considered when interpreting the results, as it could explain the fact that LepR-Cre mediated deletion has no impact in young adults.

Other comments:

- The panels in Figure 1A should be made bigger so the contribution of the different datasets maybe appreciated, and the populations clearly distinguished. At the size provided it is not possible to visualize rare subsets like Schwann cells. The same comment applies to the panels in Figure 1F, where myofibroblast, Schwann cells or pericyte clusters are not even visible.
- References for the phenotypic characterization of chondrocytes should be provided
- In Figure 2A and 2B it seems surprising that 65% of the LepR population is GFP+ while Cxcl12+ cells 100% are positive. Since Cxcl12-tdTom+ cells include should include all the LepR+ CARc, how is this finding explained?
- The UMAP in Figure 6A contains several unclustered cells. What do these correspond to?
- Is there a reactive expansion of naïve T cells numbers in the absence of CD8TCM?
- Some of the cell subsets described by the authors in the integration are not well described and therefore it is unclear if they do exist as such in the BM. As an example, the MSC chondro-subset was only present in the data from Baryawno et al., but the identity and function of the cells falling under this cluster is unknown to date. It seems relevant in this case given that, as far as one can tell from the graphs, at least a fraction of these cells expresses IL15

Reviewer #3

(Remarks to the Author)

This paper outlines in detail the expression of Il15 in bone marrow stromal cells and reveals unappreciated expression profiles and heterogeneity of Il15 expressing non-hematopoietic cells.

The effects of il15 deletion in specific subsets is then investigated and roles for BM stromal Il15 in supporting various IL-15R-dependent immune subsets in revealed. These effects are minor, but real, and adds to our understanding of IL-15 biology in hematopoietic cell development.

One general question is, given the increase in IL-15 potency upon trans-presentation by IL-15Ra and high concordance in co-expression of Il15 and Il15ra in myeloid cells, can the authors comment on the co-expression of il15 and il15ra in the BM

stromal subsets investigated? The scRNAseq dataset does not show a high level of co-expression and one wonders the biological relevance of Il15 in the absence of Il15ra.

Version 1:

Reviewer comments:

Reviewer #1

(Remarks to the Author)

The authors have made effort to explain the differences between their data and those of the previous paper by Abe et al. (Cell Reports 2023) and mentioned that they improved the analyzed populations of NK cell progenitors. However, an important aspect of the previous paper is that the major producers of IL-15 for NK cell development are hematopoietic cells but not nonhematopoietic niche cells and the present paper would fall a bit short of providing a substantial advance over the previous work published in Cell Reports. I would recommend the authors to identify the hematopoietic and/or nonhematopoietic populations essential to provide NK cell progenitors with IL-15 for their development.

Reviewer #2

(Remarks to the Author)

Thank you for your answers. My concerns have been mostly addressed

Reviewer #3

(Remarks to the Author)

I am satisfied with the revised manuscript

Point-by-point response

We thank the reviewers for their time and thoughtful feedback on our work. To address the reviewer's points and concerns, we have conducted additional experiments, added additional controls, and revised the results and discussion sections accordingly. We believe that the revised version has significantly improved in clarity and the additional experiments considerably strengthened our manuscript.

In summary, we made changes in the Figures 1, 2, 3, 6, S1, formerly S5, S6, and S7; conducted new experiments or re-analyzed data to generate the new Figures 2B, 3G, S1C, S5, S6B, S7 and S9C-E, adapted the Methods section and specified additional points in the Results and Discussion sections for more clarity.

We have provided a detailed point-by-point response to the reviewer's comments below highlighting the new additions and improvements of our work. All text changes in the revised manuscript are highlighted in blue color.

Reviewer #1 (Remarks to the Author):

It has been previously shown that conditional deletion of IL-15 in Lepr⁺ cell-derived stromal cells or Prx1⁺ cell-derived stromal cells, including Osterix⁺ stromal cells, Lepr⁺ stromal cells, and osteoblasts, did not affect the numbers of NK cell precursors and mature NK cells but that conditional deletion of IL-15 in hematopoietic cells markedly reduced the numbers of NK cell precursors and mature NK cells (ref. 46; Abe et al., Cell reports 2023). The authors generated novel IL-15 knockin reporter mice, in which physiological expression of IL-15 was maintained and show IL-15 expression levels in various non-hematopoietic cell populations in the bone marrow. In addition, they show that conditional deletion of IL-15 in Osterix⁺ cell-derived stromal cells decreased slightly NK cell precursors, memory CD8⁺ T cells and NKT cells but not mature NK cells and that conditional deletion of IL-15 in Lepr⁺ cell-derived stromal cells decreased slightly memory CD8⁺ T cells and mature NK cells at the age of 18-26 weeks but not 8-13 weeks. This is an important, in-depth performed, and well-written study; however, the paper would fall a bit short of providing an advance over the previous work. Most novel aspect of the paper is the data on memory CD8⁺ T cells; however, differences seen would be small and not very impressive.

We thank Reviewer #1 for their comments. Reviewer #1 raises concerns regarding the novelty of our study and we would like to take the opportunity to clarify these points, and to present additional data to demonstrate the distinct value of our study:

(a) Terminology:

Differences in terminology of NK precursor populations and technical remarks

While we understand the comparison to the previously published work by Abe et al. regarding the IL-15 cKO lines, we would like to emphasize the significant differences in the scope, methodology, and findings between our studies. It is correct that Abe et al. used Lepr-Cre and Prx1-Cre (but not Osx-Cre) lines to delete IL-15 in stromal cells, however, they mainly focused on quantifying mature NK cells in the BM, while our work addresses BM NK cell differentiation (*pre* NK1.1 expression), which also includes rNKPs, NKG2D⁺ rNKPs and stage A iNK cells.

Notably, Abe et al. use definitions for immature NK cells that might be better suited for NK cells in the periphery. According to their definitions and provided gating strategy, all quantified iNK subsets including iNK1 express DX5 (CD49b). However, according to recent literature (e.g. see Abel et al., 2018; Bernardini et al., 2014; Fathman et al., 2011, or Fig. 3A), CD49b is a marker that is expressed only very late in BM NK cell differentiation, e.g., stage D mature

NK cells, and is therefore not suitable to comprehensively assess NK cell lineage differentiation in the BM.

The CD49b⁺ NK1.1⁺ “iNK” subset from Abe et al. would correlate with our NK1.1⁺CD49b⁺CD127⁻ mNK cells that are indeed not reduced by stromal IL-15 cKO (i.e. in line with the previous literature), and are thus mainly dependent on hematopoietic-derived IL-15, as Abe et al. continue to show.

(b) Flow cytometry quality measures

Furthermore, in order to confidently detect the rare NK cell precursor subsets, we employed a series of quality measures (Fc receptor blocking; single cell, live/dead and erythrocyte/granulocyte exclusion gates), which – as far as we can tell from the Methods and Supplementary sections – are not reported by Abe et al. Having these quality measures in place is likely crucial to confidently identify these rare precursors and to keep the technical variation low. See below an excerpt of Supplementary Fig. 2 from Abe et al:

Fig S2 (E and F) from Abe et al, 2023: “The gating strategy for NK cells (E) and IL-7Rα⁺ NK cells (F).”

[REDACTED]

(c) IL-7Rα (CD127) staining:

IL-7Rα is essential to identify pre-NKPs and rNKPs, and to distinguish rNKPs from stage A iNK cells. However, the DX5-negative fraction of the NK1.1⁺CD3⁻ gate in the flow cytometry gating strategy above (Figure S2E) appears to almost entirely lack IL-7Rα, and only a minority of CD3⁺ cells appear to express IL-7Rα (Figure S2F above). We ourselves had initial difficulties obtaining reliable IL-7Rα signals in fixed/intracellular stainings and thus refrained from using the IL-7Rα antibody in intracellular stainings since the clone A7R34 used here is **not recommended for fixation**, which was reported to severely impact detection of the CD127 antigen (see e.g., <https://www.biolegend.com/fixation>). Since loss of IL-7Rα expression marks a crucial turning point in NK cell differentiation, the underestimation of IL-7Rα⁺ cells will lead to significant differences in the resulting populations analyzed.

Figure I CD127 detection by flow cytometry.

As a quality check for our IL-7Rα (surface) staining, we could confirm that most BM T cells were IL-7Rα⁺, and only a fraction of NK1.1⁺ cells was CD127⁺ while SSC-A^{hi} Lin⁺ cells containing granulocytes were negative as expected (see Figure I).

(d) cKO setup: Differences in WT/Cre breeding strategies

The fact that Abe et al. compared cKO strains to IL15^{f/KO} (heterozygous KO) controls, while we compared to IL15^{f/f} littermates, certainly also constitutes a major difference between our two studies. It would have to be further evaluated whether a heterozygous knockout itself already leads to a phenotype or a dislodgement of the NK precursor niche that might quench the moderate stromal cKO effect.

(e) rNKPs: rNKP dependence on (stromal) IL-15

rNKPs were furthermore slightly differently characterized by Abe et al., and identified (as far as reported in the paper) without employing the above-mentioned common flow cytometry quality controls. Remarkably, Abe et al continue to show that both Vav-Cre and IL15 total KO mice have no reduction in rNKPs. This is surprising, given that these cells readily express CD122. Data from our lab using IL15RA^{KO/KO} mice, however, suggest that CD122⁺ rNKPs and later early immature NK cell precursors are already partially IL-15 dependent during the steady state, and that with distance to IL-7R α downregulation the loss of NK cell lineage cells intensifies (see Figure II).

Figure II Relative quantification (percent of live) of the indicated cell types in the bone marrow of IL15^{f/f} (C57BL/6) versus IL15RA knockout mice.

Despite all efforts to characterize the earliest fully NK cell committed NK cell precursors, it has been shown that pre-NKPs and rNKPs represent a still heterogeneous population, e.g. including cells still able to commit to the helper ILC lineage (e.g. see Constantinides et al., 2015). In contrast, we could pinpoint the biggest effect of stromal IL-15 cKO to rNKPs which express NKG2D, which might represent an already slightly more advanced differentiation state preceding the downregulation of IL-7R α and thereby transition to full IL-15 dependence.

(f) Small differences: Further data to strengthen the NK precursor phenotype

However, one might still argue that the stromal contribution seems minor in terms of biological relevance since fully mature NK cell levels remained largely unchanged. We therefore asked ourselves whether the employed phenotype would be exacerbated in situations of heightened demand of IL-15-dependent proliferation. Indeed, after NK cell depletion using NK1.1 antibodies, Prx1-Cre mice failed to cope with the heightened demand of rNKPs as their supply of rNKPs stayed significantly lower compared to WT littermates (new Figure 3G).

While the lack of precursors is apparently compensated in later differentiation stages (the main proliferative burst in NK cells has been described to happen mainly after acquisition of DX5 (Kim et al., 2002)), we could not attribute this to heightened proliferation, as Ki-67⁺ cells were not significantly different between WT and Cre mice. Instead, cells slightly upregulated the anti-apoptotic protein Bcl-2 (new **Figure S5**).

Lastly, we also asked whether stromal IL-15 cKO has an effect on NK cell function and maturation. Strikingly, NK cells expressing the inhibitory receptor Klrp1 were significantly reduced in the BM of Prx1-Cre mice. Similarly, CD11b⁺CD27⁻ mature NK cells were less abundant in Prx1-Cre mice (new **Figure S5A-B**). Of note, we observed that NK cell populations dramatically differed between fresh and frozen samples, and thus only used freshly isolated BM for our analysis (see **Figure III**). This might perhaps also explain the differences to Abe et al., since their quantification seems to be greatly underestimating the CD11b⁻ fraction compared to our gating and to the literature (CD11b⁻ cells constitute 60-70% of total NK cells in the BM, e.g. see Harms et al., 2017 Fig2B; van Helden et al., 2012 Fig3).

Figure III Abundance of NK cell subsets among total NK1.1+CD3⁻ cells in the BM of IL15^{f/f} mice from freshly isolated („fresh“) or thawed („frozen“) BM.

According to our new results, we have **revised Figures 1, 2, 3, 6, S1, (old) S5 to S7** and the changed text is highlighted in blue in the revised manuscript.

References:

- Abel, A.M., Yang, C., Thakar, M.S., Malarkannan, S., 2018. Natural killer cells: Development, maturation, and clinical utilization. *Front. Immunol.* 9, 1869. <https://doi.org/10.3389/FIMMU.2018.01869>
- Bernardini, G., Benigni, G., Antonangeli, F., Ponzetta, A., Santoni, A., 2014. Multiple Levels of Chemokine Receptor Regulation in the Control of Mouse Natural Killer Cell Development. *Front. Immunol.* 5. <https://doi.org/10.3389/fimmu.2014.00044>
- Fathman, J.W., Bhattacharya, D., Inlay, M.A., Seita, J., Karsunky, H., Weissman, I.L., 2011. Identification of the earliest natural killer cell-committed progenitor in murine bone marrow. *Blood* 118, 5439–5447. <https://doi.org/10.1182/blood-2011-04-348912>
- Harms, R.Z., Creer, A.J., Lorenzo-Arteaga, K.M., Ostlund, K.R., Sarvetnick, N.E., 2017. Interleukin (IL)-18 Binding Protein Deficiency Disrupts Natural Killer Cell Maturation and Diminishes Circulating IL-18. *Front. Immunol.* 8. <https://doi.org/10.3389/fimmu.2017.01020>
- Kim, S., Iizuka, K., Kang, H.-S.P., Dokun, A., French, A.R., Greco, S., Yokoyama, W.M., 2002. In vivo developmental stages in murine natural killer cell maturation. *Nat. Immunol.* 3, 523–528. <https://doi.org/10.1038/ni796>
- van Helden, M.J.G., de Graaf, N., Boog, C.J.P., Topham, D.J., Zaiss, D.M.W., Sijts, A.J.A.M., 2012. The Bone Marrow Functions as the Central Site of Proliferation for Long-Lived NK Cells. *J. Immunol. Baltim. Md* 1950 189, 2333–2337. <https://doi.org/10.4049/jimmunol.1200008>

Reviewer #2 (Remarks to the Author):

In this manuscript, Stecher and colleagues investigate the cellular sources of IL-15 in the bone marrow (BM) and the functional contribution of different IL-15 producing cell types to NK cell development and CD8 memory T cell (CD8 TCM) homeostasis. Through integration of previously published sc-RNAseq datasets of the BM stromal compartment, the authors report expression of *Il15* transcripts primarily in various types of endothelial and mesenchymal stromal cell (MSC) populations. They generate a new reporter model in which expression of IL15-GFP is detected mostly in non-hematopoietic stromal cells, including *LepR*⁺ MSCs (also known as *Cxcl12*-abundant reticular cells-CARc), sinusoidal endothelial cells (SECs), chondrocytes and osteoprogenitor cells. To determine the functional role of these cell types in IL15-dependent regulation of hematopoiesis, the authors generate cell-type specific IL-15-deficient mouse models by crossing *IL15^{fl/fl}* mice to different Cre lines, including *Prx-1-Cre*, *LepR-Cre*, *Osx-Cre* and *Cdh5-Cre* strains. Analyses of these mouse lines lead the authors to the general conclusion that IL-15 derived from MSC subsets primarily controls both NK cell development and CD8 TCM maintenance in the BM, while only having a minor or negligible impact on the numbers of circulating cells. They also find that EC-specific deletion of IL-15 causes a decrease of mature NK cells and CD8 Tcm cells in the blood, while not having any effects on BM populations. Finally, inspection of published datasets of human BM stroma suggests that the profile of expression of IL-15 in MSC and EC subsets is largely conserved between murine and human marrow.

Dissecting the specific contribution of stromal subsets to the regulation of distinct developmental stages of hematopoiesis is of high relevance. The organization of BM into functional, spatially restricted niches suggests that secretion of the same chemokine by different cell types may serve to control different developmental processes, which maybe spatially segregated. This notion has been experimentally confirmed in the case of *Cxcl12* and *Scf*. Yet, whether IL-15 secretion by different subsets in specific niches plays different roles is unknown. Nonetheless, in its current form, this study falls short in clarifying this crucial question. Detailed comments are provided below:

- Although the selection of murine models targeting different stromal compartments seems suitable a priori, the results obtained in the MSC-specific strains are hard to interpret and integrate into a coherent model. A thorough characterization of the mouse strains in terms of which populations and tissue regions are targeted is lacking in the manuscript. As a prime example, the *Prx1* promoter has been previously shown to target all mesenchymal subsets with high efficiency, which also include the entirety of osteolineage cells that are targeted by the *Osx-Cre* mouse model, as well as all MSC progenitor cell populations. Based on this expected overlap, it would be anticipated that the results of deleting IL15 in the *Prx-1Cre* model would, at least phenocopy, if not exceed in magnitude, those observed in the *Osx-1Cre* line. Yet, the authors observe that the impact on NK cell development is more pronounced in *Osx-Cre* mice than in *Prx1-Cre* mice. The authors point out that osteoblasts and chondrocytes are only partially targeted in *Prx1-Cre* mice. However, this is point is incorrect, as they refer to a report in which the targeting efficiency was analyzed in *Prx1-CreERT2*, but not *Prx-1-Cre* mice, while robust evidence suggests that the use of *Prx1-Cre* mice very efficiently targets all MSC derivatives in hind limbs. To explain the discrepancies between both models, the authors should thoroughly evaluate the targeting efficiency in MSC subsets of all the mouse lines employed by flow cytometry (by crossing to a *ROSA26-tdTom* mouse line) and results should be validated via imaging, which would also provide some spatial insight that could reveal the specific niches in which the cytokine is produced (and depleted in Cre lines) and therefore guide the interpretation of the results here obtained. This in-depth characterization should also

comprise stromal cells, to make sure that Il15 deletion is not affecting numbers and frequencies of relevant stromal subsets.

(a) Characterization of Osx vs Prx1 stromal cKO models

While both the Prx1-Cre and Osx-GFP-Cre models have been used extensively in many previous studies, we agree that the discrepancy in phenotypes (concerning CD8 T_{RM} and NKT cells) is somewhat unexpected. Although a complete side-by-side characterization of all the Cre-targeted cells and their locations is not something that we will be able to accomplish within the three months allowed for manuscript revision, we conducted several experiments, re-analyzed data, and added additional controls in order to better characterize the differences between Prx1 and Osx-mediated IL-15 cKO.

We considered the following possibilities:

- (1) The additional effects in the Osx-Cre strain concerning CD8 T_{RM} and NKT cells could be due to off-target effects, e.g. deletion in hematopoietic cells or a “Cre only” transgene effect.
- (2) Prx1-Cre and Osx-Cre models might (despite the notion that non-inducible Prx1-Cre should target all hind limb mesenchymal cells and thereby in theory include all Osx-Cre targeted cells) show differences in the cell populations that they target.
- (3) IL-15 cKO might have a differential effect on the stromal cells themselves, possibly with a synergistic effect of the Osx-GFP-Cre transgene.

To address the **first point**, we first asked whether the additional Osx-Cre immune cell reduction was an effect originating from expression of the Osx-GFP-Cre transgene. In addition to CD8 T_{RM} cells, we also found that NKT cells were not reduced in “Cre only” controls (new Figure S6B), suggesting that the reduction of these cell types in Il15fl Osx-Cre mice depends on the deletion of IL-15.

Next, we considered whether Osx-Cre might delete IL-15 in hematopoietic cells, which have been shown to be important for providing IL-15 to the memory CD8 T cell compartment. We considered this an especially important control since Osx-Cre (albeit using a different Cre line, in the context of tumors) was reported to mark CD45⁺ cells in one study (Ricci et al., 2020).

However, in this context it has to be considered that some stromal Cre Tomato^{dim} cells that are CD45⁺ or CD31⁺ are typically detectable by flow cytometry in digested bone (in our hands at 1-2 logs less fluorescence intensity, e.g. see also literature examples in Figure IV for Lepr-Cre and Prx1-Cre).

[REDACTED]

Figure IV Literature examples on tdTomato^{dim} background in CD45⁺/CD31⁺ cells (Gao et al., 2023; Greenbaum et al., 2013; Matsushita et al., 2023)

The fact that these populations are not detectably Tomato⁺ by immunohistochemistry, and control experiments which have suggested that there is no significant deletion in these “off-target” populations (Greenbaum et al., 2013) would argue against off-target effects by Tomato^{dim} hematopoietic cells. We speculate that this CD45/CD31⁺ population might represent cells sticking to extracellular tdTomato fluorophore freed during organ digestion, or

phagocytic cells that take up Tomato⁺ debris, also since in our hands these cells are much less abundant in undigested BM.

Still, we found that digested BM and bone of *Osx*-Tomato mice contained a considerable amount of CD45⁺ Tomato^{dim} cells (new Figure S7A) which might hint at hematopoietic off-target cutting even when taking the aforementioned points into consideration. Most of these Tomato^{dim} cells were of myeloid origin, expressing Ly6G and/or Ly6C. Therefore, we next sorted or MACS-enriched different immune cell populations from IL15fl *Osx*-Cre mice and WT littermates and determined IL-15 expression by qPCR. Neither CD8 T cells, NK cells, Ly6G⁺ SSC^{hi} granulocytes nor macrophages had a detectable reduction of IL-15 transcripts as determined by qPCR (Figure S7B).

There remains the possibility that an unknown rare subpopulation of *Osx*-Tomato⁺ hematopoietic cells might be responsible for the *Osx*-Cre specific effects, however, the notion that the phenotype is bone-specific, and that there was no significant overall reduction in IL-15 protein levels in the BM would again not be a strong indicator to support this.

Finally, macrophage efferocytosis of apoptotic osteoblasts, which is an important process during bone homeostasis (Xu et al., 2023) might at least in part account for these Tomato^{dim} cells.

To address the **second point**, we first reviewed the literature in order to find possible differences in the reported cell populations targeted by the two Cre lines. As Reviewer #2 correctly pointed out, several aforementioned studies used an inducible Prx1-Cre line, which would differ in its targeting capability compared to non-inducible Cre that would already be active during hind limb development. On the other hand, it has to be mentioned that (non-inducible) *Osx*-Cre has also been shown to mark the stroma during development, including perivascular stromal cells / Cxcl12 abundant cells (Boulais and Frenette, 2015; Greenbaum et al., 2013; Liu et al., 2013) and is thus very likely not an exclusive “osteolineage” line and might target many similar precursor populations as Prx1-Cre. Similarly, we have found that the *Osx*-Tomato population contains a fraction of Lepr⁺Vcam1⁺ cells and also marks MSCs in addition to osteoblasts, osteocytes and trabecular bone.

However, there are some instances when *Osx*-Cre targets cells that are not marked by (non-inducible) Prx1-Cre: While Prx1-Cre apparently deletes very poorly in the calvaria and ribs (Ni et al., 2025), *Osx*-Cre marks stroma in these bones (Atsawasawan et al., 2017; W. Chen et al., 2014; Maes et al., 2010). Bone marrow outside of the long bones constitutes an important source of memory CD8 T cells (Geerman et al., 2016), although it admittedly remains unclear how hind limb residency could be affected by changes in the periphery. *Osx*-Cre has been shown to target hypertrophic chondrocytes and perivascular smooth muscle cells in addition to osteoblasts and osteocytes (J. Chen et al., 2014), and examples from the literature using *Osx*-Tomato or Prx1-Tomato lines suggest that *Osx*-Cre is more efficient in labelling trabecular bone and endosteal osteoblasts, whereas Prx1-Cre has a less pronounced Tomato signal in endosteal osteoblast but also labels the periosteum (see Figure V).

[REDACTED]

Figure V Literature examples showing Cre fate reporter lines in mouse femurs. Chen: Osx-Cre R26-mT/mG mice in femur of a 2 month old mouse with details showing labelled osteoblasts and osteocytes in cortical bone and trabecular bone. Yang: IF pictures comparing femurs from 6 week old LepR-Tomato Runx2GFP and tamoxifen-induced iOsx-Tomato Runx2GFP mice. Asterisks mark mature bone-lining osteoblasts. Stetsiv: Femur of a Prx1-Cre Tomato reporter mouse with labelled periosteum (Tb = trabecular bone). Bottom: IF pictures of Prx1-Tomato and Osx-Tomato femurs with view of the BM and cortical bone/periosteum (J. Chen et al., 2014; Stetsiv et al., 2024; Yang et al., 2017).

In a direct comparison, Osx-Cre-mediated deletion of Cxcl12 seemed to have a greater effect in mobilizing hematopoietic stem cells to the blood compared to Prx1-Cre, despite being dispensable for HSC maintenance (Greenbaum et al., 2013). Similar to their importance in HSC retention, Osx-Cre targeted stromal cells might play an especially important role in the retention of CD8 T_{RM} cells in the BM.

We then analyzed flow cytometry data from Prx1-Tomato and Osx-Tomato mice to investigate differential fate-mapping of these lines. Unsurprisingly, Prx-Tomato marked a higher amount of cells compared to Osx-Cre. Osx-Cre predominantly targeted stromal cells digested from bone chips, while Prx1-Tomato cells were also abundant in flushed and digested bone marrow, and Osx-Tomato cells had a higher fraction of CD24-expressing cells (new Figure S7C-D). However, 30-50% of the Lin⁻CD45⁻CD31⁻ stromal fraction still remained Tomato-

negative even in the Prx1-Cre mice, which at least in part might contain hematopoietic precursors (Boulais et al., 2018).

To address the **third point**, we digested stromal cells from bone of IL15^{flox} Osx-Cre and IL15^{flox} Prx1-Cre mice to determine whether IL15 cKO had an effect on abundance and functional markers of the stromal cells themselves.

Strikingly, we observed changes in the stromal marker landscape between Osx-Cre – but not Prx1-Cre – mice and IL15^{flox/flox} controls. There was a trend of reduced Cd1d, Vcam1 and Cd200 expressers among Osx-Cre mice (normalized to WT controls to account for inter-experimental variation), indicating that the IL-15 knockout has a phenotypic effect on Osx-Cre targeted stromal cells themselves (new Figure S7E). In line with this, IL15Ra has been shown to play a role in bone mineralization, and Cd200 was reduced in differentiation cultures of IL15RA KO mice *in vitro* (Loro et al., 2017).

Of major importance, it would be key to understand how targeting of the different populations affects the overall levels of IL-15 and IL-15RA in BM supernatants, which may be measured using ELISA techniques.

(b) Response: IL-15 ELISA

Overall IL15 levels from total bone marrow lysates were not significantly reduced in any of the stromal cKOs as determined by ELISA (see new Figure S9E), which further underlines that it is unlikely that the differential Osx-GFP-Cre effect stems from hematopoietic off-target effects.

• A previous publication by Abe et al. (2023 Cell Reports 42, 113127) had already reported a lack of significant defects in NK cell development in LepR-Cre Il15f/f and Prx-1Cre Il-15f/f mice. How do the authors explain the discrepancies with these reports (especially in the case of Prx-1Cre mice)? In this same report, NK cells and progenitors are found either scattered or in clusters in BM tissues. It would be relevant to understand whether the defects found in Osx-Cre mice apply to both cellular distributions.

(c) Response: Differences to previous studies

- One of the main differences to Abe et al. is their definition of immature NKs: all quantified iNK subsets including iNK1 are gated as DX5⁺ (CD49b), which (e.g. see Abel et al., 2018; Bernardini et al., 2014; Fathman et al., 2011, or Fig. 3A) is a marker that is expressed only very late in BM NK cell differentiation, although not all DX5⁺ NK cells are functionally mature. Thus, this definition might help detect iNK in other tissues after BM egress, but is not suitable to assess NK cell differentiation pathways in the BM. Since we show that CD49b⁺ mNK are not affected, this is in agreement with the data from Abe et al., that CD49b⁺ NK cells (which they call iNK1) are not affected in Prx1- and Lepr-Cre mice.

- Abe et al. compared cKOs to IL15f/KO (heterozygous KO) mice, while we used IL15f/f littermates. It would have to be further evaluated whether a heterozygous knockout itself already leads to a phenotype or a dislodgement of the NK precursor niche that might quench the moderate stromal cKO effect.

- In contrast to Abe et al, we employed numerous flow cytometry quality controls (Fc receptor blocking, live/dead staining and doublet exclusion, erythrocyte and granulocyte exclusion gates) which we think are essential for correctly identifying the rare NK cell differentiation steps among the bulk of differentiating hematopoietic precursors. In particular, we took great care in getting a bright IL-7R α staining (which is only recommended for surface and not intracellular

staining by the manufacturer) since this marker is an essential turning point in NK cell differentiation.

- rNKPs were slightly differently characterized by Abe et al., and identified (as far as reported in the paper) without employing the above-mentioned quality controls. Surprisingly, both Vav-Cre and IL15 total KO mice showed no reduction in rNKPs, although they are defined by CD122 expression. In contrast, data from our lab using IL15RA^{KO/KO} mice suggest that CD122⁺ rNKPs and later early immature NK cell precursors are already partially IL-15 dependent during the steady state, and that with distance to IL-7R α downregulation the loss of NK cell lineage cells intensifies (see Figure VI)

Figure VI Relative quantification (percent of live cells) of the indicated cell types in the bone marrow of IL15^{ff} (C57BL/6) versus IL15RA knockout mice.

- NK cell scattering/clustering as described by Abe et al. is reported for NK cells post NK1.1 expression. We currently see no technically feasible way to determine whether rNKP/early stage iNK (which are affected by stromal IL-15) are spatially distributed since their identification warrants strict exclusion of other cell types and there is no simple set of markers to locate them.

• The effects of selectively depleting Il15 in MSC subsets, on NK cell development are relatively minor and selectively apply to progenitor stages while sparing mature NK cells. How do the authors envision that developmental defects are compensated in mature subsets?

• Most importantly, given the magnitude of the effects observed both in NK as well as CD8TCM cells it would be crucial to test the functional consequences of these minor reductions in a pathological setting when increased numbers of these cellular subsets need to be mobilized.

(d) Response: **Compensation of developmental defects – functional relevance**

As Reviewer #2 points out, the observed stromal effect on NK cell development is moderate during the steady state and seems to be compensated during later differentiation stages. We could not attribute this to heightened proliferation, as Ki-67⁺ cells were not significantly different between WT and Cre mice. Instead, cells slightly upregulated the anti-apoptotic protein Bcl-2 (**new Figure S5**). The main proliferative burst in NK cells reportedly happens mainly after acquisition of DX5 (Kim et al., 2002), which might explain the compensation of the observed reduction in NK cell precursors.

To skew the system in the direction of heightened demand of NK cell precursor differentiation, we depleted NK cells using NK1.1 antibodies. Compared to WT littermates, Prx1-Cre mice failed to cope with the upregulation of precursor cells as their supply of rNKPs stayed significantly lower (**new Figure 3G**). Thus, stromal cKO failed to upregulate the expansion of NKPs during NK cell depletion, and also resulted in a mild NK cell exhaustion such as reduced Klr1 expression. However, following *in vitro* stimulation of Il15^{fllox} and Il15^{fllox} Prx1-Cre BM

extracts cells with PMA/Iono, NK cells showed a similar capacity to produce the effector molecules Irfng and Tnfa in both groups (**new Figure S5**).

• The expression patterns of the new IL-15 reporter mouse model do not fully recapitulate those observed in previously generated mouse reporter lines. More specifically, no expression of IL-15 in ECs and osteoblasts was found in these previous reports. Also, 90% of LepR+ cells were shown to express the cytokine compared to the 65% observed here. Given these discrepancies, and considering that a large fraction of sorted GFP+ cells appear not to express Il15 transcripts in the sc-RNAseq data (Figure 1F), the authors should carefully assess using qRT-PCR, to what extent GFP expression reflects that of IL-15 in their mouse model in all of the cell types studied.

(e) Response: IL-15 expression patterns

It is true that a large fraction of stromal cells, even in IL-15^{GFP} sorted cells, do not appear to express IL-15 according to scRNA-seq. In contrast, sorting of IL-15^{GFP-} and IL-15^{GFP+} MSCs, but also macrophages (**new Figure S1C**) faithfully represents IL-15 expression when determined by qPCR. As briefly discussed in the manuscript, scRNA-seq data – unlike traditional RNA-seq data – are heavily biased toward underestimating weakly expressed genes (Hicks et al., 2018). Since both IL-15 and IL-15ra are weakly expressed and transcriptionally tightly

Figure VII – Frequency of cells expressing (with a value >0) Il15 and/or Il15ra. scRNA-seq data from all myeloid cell subsets annotated in the Bone Marrow Tabula Muris dataset.

regulated transcripts, they might not be adequately detected using this method. As a reference of known IL-15 producers, we therefore downloaded scRNA-seq data from the bone marrow myeloid subsets of the public *Tabula Muris* project (Schaum et al., 2018) and queried it for IL-15/IL-15ra expression. Strikingly, most BM myeloid cells would appear to be negative for both IL-15 and IL-15ra (**Figure VII**), although myeloid cells such as monocytes are generally acknowledged to be presenting IL-15 and biologically relevant for the survival and proliferation of IL-15 dependent immune cells. We concluded that IL-15 expression cannot be faithfully quantified by scRNA-seq, at least in unstimulated cells.

As an additional quality control of our reporter model, we quantified the IL-15 transcript by qPCR in sorted IL-15^{GFP-} and IL-15^{GFP+} macrophages (**new Figure S1C**). We do not share the view that IL-15 detection of our reporter is in contrast with previous reports. Since our model contains a bicistronic construct that would only report when IL-15 protein is translated at the ribosome (at the same rate that IL-15 itself is produced, and not whenever the IL-15 promoter is active), it is not surprising that our model would result in a slightly lower percentage of positive cells compared to transgenic models. The LepR+ population appears to be shifted toward higher GFP expression compared to the fluorescence minus one control (**Fig. 2A**), but

apparently not all cells produce a high amount of IL-15 protein at all times. So perhaps while providing a more physiological representation of IL-15 protein expression, our model might suffer from a weaker reporter signal compared to other models (which can be a disadvantage depending on the technical method used).

Cui et al. (2014) have shown IL-15 expression in blood endothelial cells of multiple organs; for the BM they negatively enriched and did not differentiate between (Sca1^{dim} Icam1⁺) sinusoidal and (Sca1^{hi} Icam1⁻) arteriolar EC which might explain this discrepancy (similarly, we did not detect any GFP expression in arteriolar EC). When sorting endothelial cells from spleens of tamoxifen-induced IL15^{fllox} Cdh5-Cre/ERT mice, a reduction of IL-15 expression in Cre versus WT mice was detected (new Figure S9D).

- The mouse models employed to target MSCs constitutively express the Cre transgene, and therefore expression of Il15 is targeted throughout development. Therefore, some of the effects observed could be derived from the impact of IL-15 deletion in NK progenitors at these early stages. This point should be considered when interpreting the results, as it could explain the fact that LepR-Cre mediated deletion has no impact in young adults.

(f) Response: Cre off-target effects in immune cells

This is a very important control to be considered, please see our response in point (a) and new **Figure S7b**, which shows IL-15 monitoring in multiple immune cell subsets including NK cells (which should also have detectably lower IL-15 mRNA if IL-15 was deleted in NK progenitors).

Other comments:

- The panels in Figure 1A should be made bigger so the contribution of the different datasets maybe appreciated, and the populations clearly distinguished. At the size provided it is not possible to visualize rare subsets like Schwann cells. The same comment applies to the panels in Figure 1F, where myofibroblast, Schwann cells or pericyte clusters are not even visible.

(g) Response: Formatting

We have adjusted Figure 1A, 1B, 1F, 1G, 6B and S1E in order to better visualize expression of the Feature plots. We also changed the formatting and added zoom-in pictures for Fig 1A, 1F, and S1E in order to better visualize small clusters containing few cells. We hope that this solution allows interpreting the data adequately.

- References for the phenotypic characterization of chondrocytes should be provided

(h) Response: Reference

We have mentioned a reference on CD200+ chondrocytes in the discussion section. To make this clearer, additional chondrocyte gating references have been added to the methods/Figure S2 Figure legend (Bell et al., 2022; Belluoccio et al., 2010). CD24 is similarly used for flow cytometric analysis of human chondrocytes (Grandi et al., 2020; Lee et al., 2016). Both CD200 and CD24 are not exclusive to stromal cells so there remains the caveat of co-detecting hematopoietic cells that were not excluded by the quality and lineage gates. However, in non-digested BM samples, a stromal CD200⁺CD24⁺ chondrocyte population is virtually undetectable, providing little support for the caveat mentioned above.

- In Figure 2A and 2B it seems surprising that 65% of the LepR population is GFP+ while Cxcl12+ cells 100% are positive. Since Cxcl12-tdTom+ cells include should include all the LepR+ CARc, how is this finding explained?

(i) Response: **Cxcl12 vs Lepr IL-15 positivity**

We have added a quantification to **new Figure 2B** which shows that most, but not all, Cxcl12-DsRed+ stromal cells are also GFP+. Since our GFP reporter translates GFP only when IL-15 is translated, it might underestimate the IL-15 (transcript) expressers on a population basis compared to transgenic models. On the other hand, the Cxcl12 reporter has a very bright fluorophore that might overestimate GFP+ cells for technical reasons because it is hard to compensate these channels which might explain the remaining discrepancy. We also have to consider that the Cxcl12-DsRed strain is a knock-in/knock-out reporter where one Cxcl12 allele is replaced by the fluorophore, thus these mice are effectively heterozygous knockouts for Cxcl12 and might therefore not be directly comparable to IL15-GFP single reporters or tdTomato reporters.

- The UMAP in Figure 6A contains several unclustered cells. What do these correspond to?

(j) Response: **Extended quality control on human public dataset**

For the human dataset, the stromal cell subset was used as annotated in the original publication, resulting in some unclustered speckles when showing the data without overwriting the Umap coordinates pinned to the Seurat object by the authors. We therefore re-did the Umap projection and new unsupervised clustering of the stromal subset, including the implementation of additional quality controls (removal of *Ptpnc* and/or *Gypa* expressing clusters) which led to the conclusion that the unclustered come from contaminating (*Ptpnc*+, *CD79*+) hematopoietic cells, which we then removed in an additional quality control step. The new Figure 6A now contains the cleaned Umap plot with clusters very similar to the original publication. Figures 6B-D were adjusted accordingly.

- Is there a reactive expansion of naïve T cells numbers in the absence of CD8TCM?

(k) Response: **Reactive naïve T cell expansion**

Re-analysis of our cKO datasets suggests that naïve CD8 T cells were not reactively expanded in the bone marrow of any of the cKO strains tested (see Figure VIII).

Figure VIII Naive CD8 T cell percentages (top) and numbers (bottom) in BM of the indicated IL-15 cKO strains (Cdh5-WT and Cdh5-Cre/ERT2 mice were induced with Tamoxifen at 7-10 weeks of age). P values from unpaired t-tests are indicated above the respective plot.

• Some of the cell subsets described by the authors in the integration are not well described and therefore it is unclear if they do exist as such in the BM. As an example, the MSC chondro-subset was only present in the data from Baryawno et al., but the identity and function of the cells falling under this cluster is unknown to date. It seems relevant in this case given that, as far as one can tell from the graphs, at least a fraction of these cells expresses IL15

(I) Response: Characterization of the MSC_chondro subset

It is true that most (but not all) of the cells from the “MSC-chondro” cluster stem from the Baryawno dataset, and we fell short in characterizing this cell type. By splitting the dataset by individual batches, it becomes clear that the MSC_chondro cluster stems particularly from the batches derived from digested bone (Figure IX).

Figure IX Frequency of “MSC-chondro” clustered cells within the individual batches of the public datasets used (left). Right: ViolinPlot of chondrocyte markers in the MSC_stem versus the MSC_chondro cluster.

Baryawno et al. contains the biggest dataset, and particularly the largest number of cells derived from digested bone chips. Furthermore, Baryawno et al. - in contrast to all other studies - used a slightly different digestion protocol using 1 mg/mL Dispase and 1 mg/mL STEMxyme, which also contains neutral protease and caseinase. Therefore, the simplest explanation is that this is a particularly rare and/or hard to digest cell type found primarily in the bone fraction. From a quality control perspective, we saw no reason to exclude this cluster. The MSC_chondro cluster is transcriptionally very similar to “MSC_stem”, but was identified as a distinct entity by the clustering algorithm. The cluster also showed a normal number of RNA features suggesting that it is likely not composed of doublets. In contrast to the MSC_stem cluster, MSC_chondro cells expressed the chondrocyte markers Col9a2, Col2a1 and Acan (Figure IX, right panel).

References:

- Abel, A.M., Yang, C., Thakar, M.S., Malarkannan, S., 2018. Natural killer cells: Development, maturation, and clinical utilization. *Front. Immunol.* 9, 1869. <https://doi.org/10.3389/FIMMU.2018.01869/BIBTEX>
- Atsawasuwan, P., Oubaidin, M., Dalal, B., Khan, H., Arshad, M., 2017. Calvarial bone development and suture closure in Dicer deficient mice. *Orthod. Craniofac. Res.* 20, 26–31. <https://doi.org/10.1111/ocr.12169>
- Bell, N., Bhagat, S., Muruganandan, S., Kim, R., Ho, K., Pierce, R., Kozhemyakina, E., Lassar, A.B., Gamer, L., Rosen, V., Ionescu, A.M., 2022. Overexpression of transcription factor FoxA2 in the developing skeleton causes an enlargement of the cartilage hypertrophic zone, but it does not trigger ectopic differentiation in immature chondrocytes. *Bone* 160, 116418. <https://doi.org/10.1016/j.bone.2022.116418>
- Belluoccio, D., Etich, J., Rosenbaum, S., Frie, C., Grskovic, I., Stermann, J., Ehlen, H., Vogel, S., Zaucke, F., Mark, K. von der, Bateman, J.F., Brachvogel, B., 2010. Sorting of growth plate chondrocytes allows the isolation and characterization of cells of a defined differentiation status. *J. Bone Miner. Res.* 25, 1267–1281. <https://doi.org/10.1002/jbmr.30>
- Bernardini, G., Benigni, G., Antonangeli, F., Ponzetta, A., Santoni, A., 2014. Multiple Levels of Chemokine Receptor Regulation in the Control of Mouse Natural Killer Cell Development. *Front. Immunol.* 5. <https://doi.org/10.3389/fimmu.2014.00044>
- Boulais, P.E., Frenette, P.S., 2015. Making sense of hematopoietic stem cell niches. *Blood* 125, 2621. <https://doi.org/10.1182/blood-2014-09-570192>
- Boulais, P.E., Mizoguchi, T., Zimmerman, S., Nakahara, F., Vivié, J., Mar, J.C., van Oudenaarden, A., Frenette, P.S., 2018. The Majority of CD45⁺ Ter119⁺ CD31⁺ Bone Marrow Cell Fraction Is of Hematopoietic Origin and Contains Erythroid and Lymphoid Progenitors. *Immunity* 49, 627-639.e6. <https://doi.org/10.1016/j.immuni.2018.08.019>
- Cao, J., Jin, L., Yan, Z.-Q., Wang, X.-K., Li, Y.-Y., Wang, Z., Liu, Y.-W., Li, H.-M., Guan, Z., He, Z.-H., Gong, J.-S., Liu, J.-H., Yin, H., Tan, Y.-J., Hong, C.-G., Feng, S.-K., Zhang, Y., Wang, Y.-Y., Qi, L.-Y., Chen, C.-Y., Liu, Z.-Z., Wang, Z.-X., Xie, H., 2023. Reassessing endothelial-to-mesenchymal transition in mouse bone marrow: insights from lineage tracing models. *Nat. Commun.* 14, 8461. <https://doi.org/10.1038/s41467-023-44312-w>
- Chen, J., Shi, Y., Regan, J., Karuppaiah, K., Ornitz, D.M., Long, F., 2014. Osx-Cre targets multiple cell types besides osteoblast lineage in postnatal mice. *PLoS One* 9, e85161. <https://doi.org/10.1371/journal.pone.0085161>
- Chen, W., Ma, J., Zhu, G., Jules, J., Wu, M., McConnell, M., Tian, F., Paulson, C., Zhou, X., Wang, L., Li, Y.-P., 2014. Cbfb deletion in mice recapitulates cleidocranial dysplasia and reveals multiple functions of Cbfb required for skeletal development. *Proc. Natl. Acad. Sci.* 111, 8482–8487. <https://doi.org/10.1073/pnas.1310617111>
- Cui, G., Hara, T., Simmons, S., Wagatsuma, K., Abe, A., Miyachi, H., Kitano, S., Ishii, M., Tani-ichi, S., Ikuta, K., 2014. Characterization of the IL-15 niche in primary and secondary lymphoid organs in vivo. *Proc. Natl. Acad. Sci.* 111, 1915–1920. <https://doi.org/10.1073/pnas.1318281111>
- Fathman, J.W., Bhattacharya, D., Inlay, M.A., Seita, J., Karsunky, H., Weissman, I.L., 2011. Identification of the earliest natural killer cell-committed progenitor in murine bone marrow. *Blood* 118, 5439–5447. <https://doi.org/10.1182/blood-2011-04-348912>
- Geerman, S., Hickson, S., Brasser, G., Pascutti, M.F., Nolte, M.A., 2016. Quantitative and Qualitative Analysis of Bone Marrow CD8⁺ T Cells from Different Bones Uncovers a Major Contribution of the Bone Marrow in the Vertebrae. *Front. Immunol.* 6, 660. <https://doi.org/10.3389/fimmu.2015.00660>
- Grandi, F.C., Baskar, R., Smeriglio, P., Murkherjee, S., Indelli, P.F., Amanatullah, D.F., Goodman, S., Chu, C., Bendall, S., Bhutani, N., 2020. Single-cell mass cytometry reveals cross-talk between inflammation-dampening and inflammation-amplifying cells in osteoarthritic cartilage. *Sci. Adv.* 6, eaay5352. <https://doi.org/10.1126/sciadv.aay5352>

- Greenbaum, A., Hsu, Y.M.S., Day, R.B., Schuetzpelz, L.G., Christopher, M.J., Borgerding, J.N., Nagasawa, T., Link, D.C., 2013. CXCL12 in early mesenchymal progenitors is required for haematopoietic stem-cell maintenance. *Nat.* 2013 4957440 495, 227–230. <https://doi.org/10.1038/nature11926>
- Hicks, S.C., Townes, F.W., Teng, M., Irizarry, R.A., 2018. Missing data and technical variability in single-cell RNA-sequencing experiments. *Biostat. Oxf. Engl.* 19, 562–578. <https://doi.org/10.1093/biostatistics/kxx053>
- Kim, S., Iizuka, K., Kang, H.-S.P., Dokun, A., French, A.R., Greco, S., Yokoyama, W.M., 2002. In vivo developmental stages in murine natural killer cell maturation. *Nat. Immunol.* 3, 523–528. <https://doi.org/10.1038/ni796>
- Lee, J., Smeriglio, P., Drago, J., Maloney, W.J., Bhutani, N., 2016. CD24 enrichment protects while its loss increases susceptibility of juvenile chondrocytes towards inflammation. *Arthritis Res. Ther.* 18, 292. <https://doi.org/10.1186/s13075-016-1183-y>
- Liu, Y., Strecker, S., Wang, L., Kronenberg, M.S., Wang, W., Rowe, D.W., Maye, P., 2013. Osterix-Cre Labeled Progenitor Cells Contribute to the Formation and Maintenance of the Bone Marrow Stroma. *PLOS ONE* 8, e71318. <https://doi.org/10.1371/journal.pone.0071318>
- Loro, E., Ramaswamy, G., Chandra, A., Tseng, W.-J., Mishra, M.K., Shore, E.M., Khurana, T.S., 2017. IL15RA is required for osteoblast function and bone mineralization. *Bone* 103, 20–30. <https://doi.org/10.1016/j.bone.2017.06.003>
- Maes, C., Kobayashi, T., Selig, M.K., Torrekens, S., Roth, S.I., Mackem, S., Carmeliet, G., Kronenberg, H.M., 2010. Osteoblast Precursors, but Not Mature Osteoblasts, Move into Developing and Fractured Bones along with Invading Blood Vessels. *Dev. Cell* 19, 329–344. <https://doi.org/10.1016/j.devcel.2010.07.010>
- Matsushita, Y., Liu, J., Chu, A.K.Y., Tsutsumi-Arai, C., Nagata, M., Arai, Y., Ono, W., Yamamoto, K., Saunders, T.L., Welch, J.D., Ono, N., 2023. Bone marrow endosteal stem cells dictate active osteogenesis and aggressive tumorigenesis. *Nat. Commun.* 14, 2383. <https://doi.org/10.1038/s41467-023-38034-2>
- Ni, Y., Wu, J., Liu, F., Yi, Y., Meng, X., Gao, X., Xiao, L., Zhou, W., Chen, Z., Chu, P., Xing, D., Yuan, Y., Ding, D., Shen, G., Yang, M., Wu, R., Wang, L., Melo, L.M.N., Lin, S., Cheng, X., Li, G., Tasdogan, A., Ubellacker, J.M., Zhao, H., Fang, S., Shen, B., 2025. Deep imaging of LepR+ stromal cells in optically cleared murine bone hemisections. *Bone Res.* 13, 1–17. <https://doi.org/10.1038/s41413-024-00387-9>
- Ricci, B., Tycksen, E., Celik, H., Belle, J.I., Fontana, F., Civitelli, R., Faccio, R., 2020. Osterix-cre marks distinct subsets of CD45- and CD45+ stromal populations in extra-skeletal tumors with pro-tumorigenic characteristics. *eLife* 9, 1–29. <https://doi.org/10.7554/ELIFE.54659>
- Schaum, N., Karkanias, et al., The Tabula Muris Consortium, Overall coordination, Logistical coordination, Organ collection and processing, Library preparation and sequencing, Computational data analysis, Cell type annotation, Writing group, Supplemental text writing group, Principal investigators, 2018. Single-cell transcriptomics of 20 mouse organs creates a Tabula Muris. *Nature* 562, 367–372. <https://doi.org/10.1038/s41586-018-0590-4>
- Stetsiv, M., Wan, M., Prabhu, S., Guzzo, R., Sanjay, A., 2024. Improved Methodology for Studying Postnatal Osteogenesis via Intramembranous Ossification in a Murine Bone Marrow Injury Model. <https://doi.org/10.1101/2024.10.24.620082>
- Xu, R., Xie, H., Shen, X., Huang, J., Zhang, H., Fu, Y., Zhang, P., Guo, S., Wang, D., Li, S., Zheng, K., Sun, W., Liu, L., Cheng, J., Jiang, H., 2023. Impaired Efferocytosis Enables Apoptotic Osteoblasts to Escape Osteoimmune Surveillance During Aging. *Adv. Sci.* 10, 2303946. <https://doi.org/10.1002/advs.202303946>
- Yang, M., Arai, A., Udagawa, N., Hiraga, T., Lijuan, Z., Ito, S., Komori, T., Moriishi, T., Matsuo, K., Shimoda, K., Zahalka, A.H., Kobayashi, Y., Takahashi, N., Mizoguchi, T., 2017. Osteogenic Factor Runx2 Marks a Subset of Leptin Receptor-Positive Cells that Sit Atop the Bone Marrow Stromal Cell Hierarchy. *Sci. Rep.* 7, 4928. <https://doi.org/10.1038/s41598-017-05401-1>

Reviewer #3 (Remarks to the Author):

This paper outlines in detail the expression of Il15 in bone marrow stromal cells and reveals unappreciated expression profiles and heterogeneity of Il15 expressing non-hematopoietic cells.

The effects of il15 deletion in specific subsets is then investigated and roles for BM stromal Il15 in supporting various IL-15R-dependent immune subsets is revealed. These effects are minor, but real, and adds to our understanding of IL-15 biology in hematopoietic cell development.

One general question is, given the increase in IL-15 potency upon trans-presentation by IL-15Ra and high concordance in co-expression of Il15 and Il15ra in myeloid cells, can the authors comment on the co-expression of il15 and il15ra in the BM stromal subsets investigated? The scRNAseq dataset does not show a high level of co-expression and one wonders the biological relevance of Il15 in the absence of Il15ra.

We have indeed wondered about the discrepancy between IL-15 and IL-15ra expression in the public scRNA-seq datasets as well as our own IL15-GFP⁺ sorted dataset. It is generally thought and has been shown in different studies using KO strains and bone marrow chimeras that IL-15 is *trans-presented*, although there are also some examples in the literature that hint at *in cis* presentation of IL-15 (mostly from *in vitro* data).

However, despite this early-established “dogma”, several other studies hint at a more complex system of IL-15 signaling, that includes e.g. shedding of soluble IL-15ra as an anti-inflammatory scavenging molecule (Bouchaud et al., 2013), trans-endocytosis of IL-15/IL-15ra complex to receiving cells (Anton et al., 2020), or an *in cis* effect of IL-15 on stromal cells that express only IL-15ra but not IL2rb/IL2rg (Kornsuthisopon et al., 2021)

Intriguingly, it has been shown that inhibition of the trimeric Il15ra/IL-2Rβ/γc receptor without affecting signaling through trans-presentation via the dimeric IL-2Rβ/γc receptor does not affect homeostatic NK and CD8 T cell numbers (Meghnam et al., 2022), implying that *in cis* signaling on immune cells is not the primary mode of action during the steady state effects of IL-15.

Still, the observation that IL-15ra is almost absent from MSCs stems from scRNA-seq data, which – unlike traditional RNA-seq data – are heavily biased toward underestimating weakly expressed genes (Hicks et al., 2018). Since both IL-15 and IL-15ra are weakly expressed transcripts, they might not be adequately detected.

As a kind of “positive control” reference, we therefore downloaded scRNA-seq data from the bone marrow myeloid subsets of the public *Tabula Muris* project

Figure X – Frequency of cells expressing (with a value >0) Il15 and/or Il15ra. scRNA-seq data from all myeloid cell subsets annotated in the Bone Marrow *Tabula Muris* dataset.

(Schaum et al., 2018) and queried it for IL-15/IL-15ra expression.

Strikingly, most BM myeloid cells would appear to be negative for both IL-15 and IL-15ra (Figure X), and the IL-15ra transcript in particular could be detected only in a few percent of cells, although myeloid cells such as monocytes are generally acknowledged to be trans-presenting IL-15/IL-15ra as this has been shown to be biologically relevant for the survival and proliferation of IL-15-dependent immune cells.

We concluded that IL-15ra expression cannot be adequately quantified by scRNA-seq. Using qPCR, IL-15ra mRNA was indeed slightly enriched in IL15-GFP⁺ versus IL15-GFP⁻ MSCs (new Fig S1A), although not as strongly as in IL15-GFP⁻ vs. IL15-GFP⁺ sorted macrophages (new Fig S1C).

IL-15ra is known to be heavily post-translationally regulated via trans-endosomal recycling rounds (Dubois et al., 2002). Additionally, the detection under steady state conditions in the absence of overexpression or strong inflammatory stimulation might be technically challenging.

We of course then also attempted to include IL-15ra into our flow cytometry panels to have a protein-based readout to be able to check IL-15/IL-15ra expression on a single cell basis. However, during our internal quality assessment using three different anti-mouse IL-15ra antibodies from three different companies, we could (1) not get a detectable positive signal in IL-15 producers such as splenic dendritic cells during the steady state, and (2) detected unspecific staining for all the used antibodies not only in WT mice, but also in IL15ra-KO mice. Since IL15ra is readily internalized (Dubois et al., 2002), we also tried intracellular staining after fixation/permeabilization, but with similar results (Figure XI).

Therefore, we concluded that using flow antibodies during the steady state is not technically feasible due to the low signal and high background.

Figure XI – Exemplary flow cytometry plots of spleen, bone marrow and bone (C75BL/6) WT and IL15RA-KO mice stained with different IL15ra antibodies.

Lastly, it is still noteworthy that IL15ra seems to be more readily detected in chondrocytes and osteoblasts based on single cell transcriptomics. The literature suggests a role of IL15ra in

bone mineralization (Loro et al., 2017), and it would be interesting to see more research as to whether these effects are directly mediated by stromal IL15ra or as a secondary consequence of the altered marrow immune cell composition. In line with this, we could observe a change in the expression of various markers in stroma from IL15fl Osx-Cre mice (see new Figure S7E).

Whether stromal cells are truly trans-presenters would eventually have to be determined via conditional deletion of IL15ra.

Please note that Figures 1, 2, 3, 6, S1, (old) S5 to S7 have been adapted/extended and the changed text is highlighted in blue in the revised manuscript.

References:

- Anton, O.M., Peterson, M.E., Hollander, M.J., Dorward, D.W., Arora, G., Traba, J., Rajagopalan, S., Snapp, E.L., Christopher Garcia, K., Waldmann, T.A., Long, E.O., 2020. Trans-endocytosis of intact IL-15R α -IL-15 complex from presenting cells into NK cells favors signaling for proliferation. *Proceedings of the National Academy of Sciences of the United States of America* 117, 522–531. <https://doi.org/10.1073/PNAS.1911678117/-/DCSUPPLEMENTAL>
- Bouchaud, G., Gehrke, S., Krieg, C., Kolios, A., Hafner, J., Navarini, A.A., French, L.E., Boyman, O., 2013. Epidermal IL-15R α acts as an endogenous antagonist of psoriasiform inflammation in mouse and man. *Journal of Experimental Medicine* 210, 2105–2117. <https://doi.org/10.1084/jem.20130291>
- Dubois, S., Mariner, J., Waldmann, T.A., Tagaya, Y., 2002. IL-15R α recycles and presents IL-15 in trans to neighboring cells. *Immunity* 17, 537–547. [https://doi.org/10.1016/S1074-7613\(02\)00429-6](https://doi.org/10.1016/S1074-7613(02)00429-6)
- Hicks, S.C., Townes, F.W., Teng, M., Irizarry, R.A., 2018. Missing data and technical variability in single-cell RNA-sequencing experiments. *Biostatistics* 19, 562–578. <https://doi.org/10.1093/biostatistics/kxx053>
- Kornsuthisophon, C., Manokawinchoke, J., Sonpoung, O., Osathanon, T., Damrongsri, D., 2021. Interleukin 15 participates in Jagged1-induced mineralization in human dental pulp cells. *Archives of Oral Biology* 128, 105163. <https://doi.org/10.1016/j.archoralbio.2021.105163>
- Loro, E., Ramaswamy, G., Chandra, A., Tseng, W.-J., Mishra, M.K., Shore, E.M., Khurana, T.S., 2017. IL15RA is required for osteoblast function and bone mineralization. *Bone* 103, 20–30. <https://doi.org/10.1016/j.bone.2017.06.003>
- Meghmem, D., Maillason, M., Barbieux, I., Morisseau, S., Keita, D., Jacques, Y., Quémener, A., Mortier, E., 2022. Selective Targeting of IL-15R α Is Sufficient to Reduce Inflammation. *Front Immunol* 13, 886213. <https://doi.org/10.3389/fimmu.2022.886213>
- Schaum, N., Karkanias, J., et al., The Tabula Muris Consortium, Overall coordination, Logistical coordination, Organ collection and processing, Library preparation and sequencing, Computational data analysis, Cell type annotation, Writing group, Supplemental text writing group, Principal investigators, 2018. Single-cell transcriptomics of 20 mouse organs creates a Tabula Muris. *Nature* 562, 367–372. <https://doi.org/10.1038/s41586-018-0590-4>

Point-by-point response

Reviewer #1 (Remarks to the Author):

The authors have made effort to explain the differences between their data and those of the previous paper by Abe et al. (Cell Reports 2023) and mentioned that they improved the analyzed populations of NK cell progenitors. However, an important aspect of the previous paper is that the major producers of IL-15 for NK cell development are hematopoietic cells but not nonhematopoietic niche cells and the present paper would fall a bit short of providing a substantial advance over the previous work published in Cell Reports. I would recommend the authors to identify the hematopoietic and/or nonhematopoietic populations essential to provide NK cell progenitors with IL-15 for their development.

Reviewer #2 (Remarks to the Author):

Thank you for your answers. My concerns have been mostly addressed

Reviewer #3 (Remarks to the Author):

I am satisfied with the revised manuscript

Author Reply

We thank the reviewers for evaluating our revised manuscript. Regarding the point that Reviewer #1 raised: As outlined in our previous, detailed point-by-point reply, Abe et al. did not analyze NK precursors in Prx1-Cre und Lepr-Cre mice, since their iNK1-2 and mNK1-2 subsets all express NK1.1, while NK precursors (rNKPs, NKG2D+ rNKPs and stage A iNK cells) do not express NK1.1. Furthermore, Abe et al. compared heterozygous to homozygous KO mice, while we compare WT to KO mice. Our results instead demonstrate that IL-15-producing Osx1+ stroma cells promote NK precursors in the BM to an extent that is similar to total IL15RA knockout mice (Figure I below). We have now included Figure IB below in our Suppl. Figures (**Figure S6D**) to highlight the contribution of Osx1+ stromal cells (Figure 4A) compared to total IL-15 KO for NK cell precursors, immature NK cells and mature NK cells in the BM. We have now mentioned this also in the results and discussion section.

Figure I: Relative quantification (% of live) of the indicated cell types in the BM of (A) IL15^{f/f} and IL15^{f/f} Osx1-Cre and (B) IL15^{f/f} (WT) and IL15RA KO mice.

Abe et al. use Vav1-Cre mice to analyze the contribution of hematopoietic cell-derived IL-15. However, in 30% of mice, Vav1-cre activity is found in non-hematopoietic as well as hematopoietic cells¹ (<https://www.jax.org/strain/035670>), and recombination also occurs in most endothelial cells². In fact, the results below from Abe et al. show that iNK1 cells, corresponding to stage B-C iNK / mNK cells, are severely affected in IL15 KO mice but show only a minor reduction in IL15 flox Vav1-Cre. This further supports our results of stromal cells being the major source for rNKP, NKG2D+ rNKP and stage A iNK cells, but not for mNK cell survival, as shown in Figure IA (Figure 4A of the manuscript).

[REDACTED]

Figure II: Graphs from Abe et al., Cell Reports, 2023

Importantly, our study goes far beyond the results published by Abe et al. We demonstrate that Prx1+ and Osx1+ stromal cells as well as endothelial cells have distinct effects on all IL-15-dependent immune cell lineages, including memory CD8⁺ T cell subsets and NKT cells. We therefore redefine functional heterogeneity among MSCs (developmental and survival niches), e.g., by showing which MSC subtypes differentially regulate the development and survival of all IL-15-dependent immune cell lineages in the BM and beyond.

References

- 1 Stadtfeld, M. & Graf, T. Assessing the role of hematopoietic plasticity for endothelial and hepatocyte development by non-invasive lineage tracing. *Development* **132**, 203-213, doi:10.1242/dev.01558 (2005).
- 2 Georgiades, P. *et al.* VavCre transgenic mice: a tool for mutagenesis in hematopoietic and endothelial lineages. *Genesis* **34**, 251-256, doi:10.1002/gene.10161 (2002).